# Tensor Network Structure Search Via Canonical Dimension Tree Enumeration

## Abstract

Tensor networks provide a powerful framework for compressing multi-dimensional data. The optimal tensor network structure for a given data tensor depends on both data characteristics and specific optimality criteria, making tensor network structure search a challenging problem. Existing solutions typically rely on sampling and compressing numerous candidate structures; these procedures are computationally expensive and therefore limiting for practical applications. We address this challenge by decoupling topology enumeration from rank assignment search. We first represent the search space using canonical dimension trees, a hierarchical structure that encodes potential network topology through nested index partitions. This representation inherently rules out redundant and suboptimal topologies by construction. To mitigate the assessment bottleneck, we introduce a mechanism powered by the precomputation of a singular value map. By archiving the singular values of all feasible tensor matricizations, we transform the evaluation of any candidate dimension tree into a constraint-solving problem. This formulation yields an empirically near-optimal rank assignment via simple metadata lookups, allowing us to compute structural costs directly and bypass expensive on-the-fly tensor decompositions for all but the final selected candidate. Experimental results show that our approach accelerates the structure search by up to 10× while achieving highly competitive compression ratios, outperforming standard tensor trains and hierarchical tuckers by up to 10×, and matching or exceeding state-of-the-art structure search tools. Notably, our approach scales to larger tensors that are unattainable by prior work. Furthermore, the discovered topologies generalize well to similar data; they achieve compression ratios up to 2.4× better than tensor trains or hierarchical tuckers, while maintaining a search time of approximately 110 seconds for 6D tensors of 1–2GB disk size.

## 1 Introduction

Tensor networks have found widespread applications in machine learning (Lebedev et al., 2014; Novikov et al., 2015; Phan et al., 2020; Memmel et al., 2022; Sprangers & Vannieuwenhoven, 2023; Panagakis et al., 2021), scientific computing Richter et al. (2021); Evenbly (2017); Gray & Chan (2024); Ma et al. (2024); Ceruti et al. (2021); Rakhuba (2021); Zhang & Kileel (2023), quantum computing Verstraete et al. (2008); Bañuls (2023); Montangero (2018); Arad & Landau (2010), among many other fields, because they allow effective low-rank approximations of high-dimensional data. Over the past decade, various tensor network structures—such as tensor trains (TT) (Oseledets, 2011), Tuckers (Tucker, 1966), and hierarchical Tuckers (HT) (Grasedyck, 2010)—have been deployed. Each of these structures offers distinct advantages for specific scenarios, with no single optimal representation across problem settings. This observation brings up an important question: given a data tensor and an optimization objective, how could one efficiently determine the most suitable tensor network structure to achieve the desired goal? This question has evolved into a research topic known as tensor network structure search (TN-SS).

TN-SS has two highly inter-related components: (1) the identification of a graph where nodes correspond to tensors and edges represent shared dimensions between connecting tensors; and (2) an assessment of the compression and approximation quality of each graph. More specifically, the task of searching for optimal tensor network structures involves two subtasks: (1) topology search: identify the optimal connections

between nodes; and (2) rank search: find the optimal dimension sizes (also called ranks) for edges. While this division provides a clear framework to address the TN-SS problem, existing approaches face significant challenges in effectively solving these subtasks. We categorize these problems into two primary challenges: the combinatorial explosion of the search space, and the prohibitive computational cost of candidate evaluation.

The first challenge is the vast search space to jointly optimize tensor network topology and ranks. A large portion of prior work has simplified this problem by focusing exclusively on either topology optimization with fixed ranks (Li & Sun, 2020; Hikihara et al., 2023; Haberstich et al., 2023) or rank optimization for a fixed topology (Rai et al., 2014; Mickelin & Karaman, 2020; Sedighin et al., 2021; Yin et al., 2022; Ghadiri et al., 2023). However, such isolated optimization often fails to identify the optimal structure where topology and rank are tightly coupled. Recent advancements attempt to address both subtasks simultaneously by sampling candidate structures from the joint search space (Hashemizadeh et al., 2020; Li et al., 2022; 2023; Zeng et al., 2024; Zheng et al., 2024). However, the required sample size grows rapidly with the data tensor size, leading to a large number of candidates. The scalability issue is further exacerbated by the introduction of internal nodes, which enhances tensor network compression performance but can be inserted anywhere within tensor networks, exponentially increasing the topological search space. While greedy heuristics have been proposed to mitigate this explosion (Hashemizadeh et al., 2020), they often sacrifice global optimality for computational feasibility, leaving the discovery of near-optimal structures an open problem.

Orthogonal to the search space explosion is the high overhead associated with candidate assessment. To determine the quality of a proposed structure, one must typically check whether the given data tensor can be compressed into this structure within the prescribed error tolerance. Standard techniques, such as alternating least squares (Kolda & Bader, 2009a; Hashemizadeh et al., 2020) or gradient descent (Kolda & Hong, 2020; Li & Sun, 2020; Li et al., 2022), involve iterative and computationally expensive tensor operations. When these assessment procedures are embedded into a large search loop, where thousands of candidates must be evaluated, the total search cost becomes prohibitive. Consequently, there is a critical need for more efficient assessment proxies that can accurately rank candidates.

## 1.1 Efficient TN-SS via Canonical Dimension Tree Enumeration

In this work, we propose a framework that decouples the topology and rank search, which is fundamentally different from prior work that attempt to sample the joint space of topologies and ranks.

To achieve this decoupling, we only enumerate tensor network topologies and solve for near-optimal rank assignments. Each network topology is represented as a *canonical dimension tree* (CDT), which is a tree describing node connectivity and index placement without assigned edge weights. CDTs generalize classical dimension trees used in tree tensor networks (Falcó et al., 2021; Grasedyck, 2010) by allowing multiple free indices to be assigned to both leaf and non-leaf nodes, while strictly enforcing a canonical representation format. The structural constraints defined in CDTs naturally prune the search space, excluding duplicate enumeration and suboptimal topologies that violate valid CDT properties.

Complementing this search space of network topologies, we introduce a constraint-solving framework as the engine for both candidate assessment and rank optimization. The constraint solving method is powered by a precomputed metadata map $\Omega$ that archives the singular values of the data tensor across all possible tensor unfoldings (Kolda & Bader, 2009b). For any candidate CDT, the framework queries $\Omega$ to retrieve precomputed singular values and solves for a near-optimal rank assignment that satisfies a target error bound. The resulting ranks yield a rigorous over-approximation of the tree's optimal size, providing a performance score to guide the structural search and completely bypassing the need for explicit rank enumeration.

Synthesizing these components into a unified pipeline, the integrated workflow illustrated in Fig. 1 consists of two primary phases: dimension tree enumeration and constraint-based candidate scoring. First, the algorithm generates a collection of CDTs, each standing for a tensor network skeleton. Due to the special design of CDTs, this phase rules out suboptimal skeletons from the search space. It also significantly reduces the search space size by ignoring rank enumerations. Second, for each dimension tree, the algorithm queries the singular value map $\Omega$ to solve the rank optimization problem, yielding an estimated compression score that allows us to rank the CDTs effectively. By delaying computationally intensive tensor decompositions

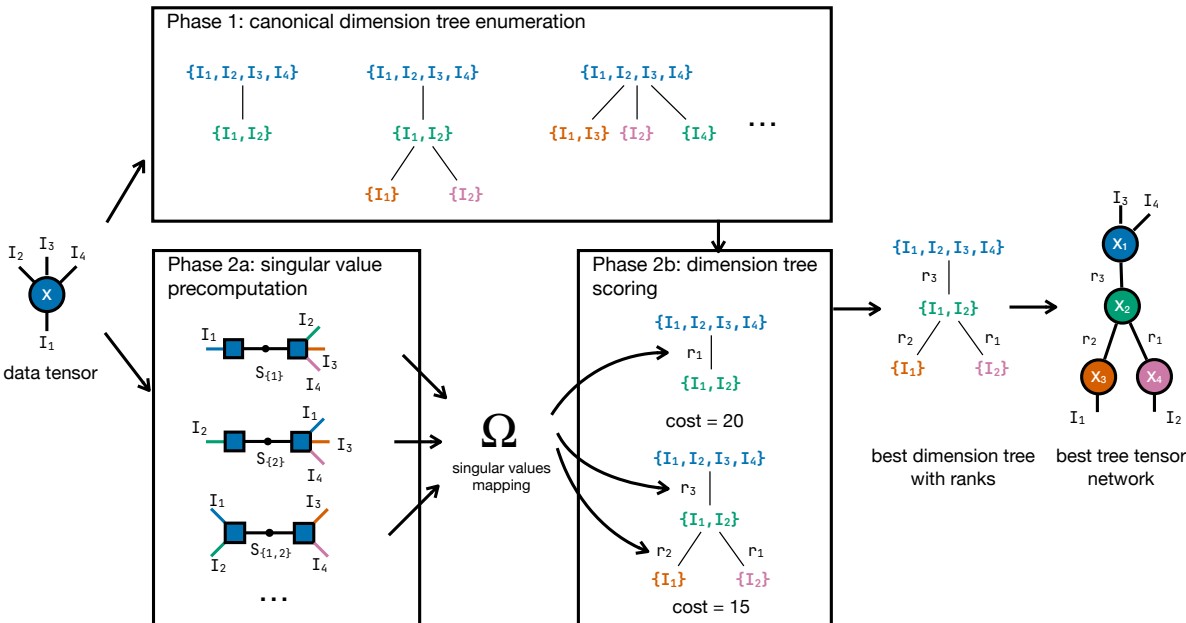

Figure 1: Overview of the proposed algorithm CADET. This algorithm consists of two phases: (1) enumeration of dimension trees, each representing a tensor network skeleton, and (2) scoring of dimension trees. The second phase includes two steps: (2a) pre-computation of singular values for all matricizations of the input tensor, and (2b) scoring of dimension trees without explicit decomposition. After the promising dimension trees are identified, tensor decomposition is run to obtain the best tree structure within the search space.

until the most promising skeletons and their ranks are determined, our strategy identifies high-performance network structures with the cost much lower than the traditional computational overhead.

**Contributions** In summary, we make the following contributions in this paper:

- We introduce canonical dimension trees to represent tensor network topologies. This representation encodes network topologies as hierarchical index partitions, and inherently rules out suboptimal and redundant skeletons by construction, effectively regularizing the search space (Section 3.2).

- We propose a novel evaluation mechanism that bypasses the need for full tensor compressions during the search phase. By formulating rank assignment as a constraint-solving problem over precomputed singular values, we can simultaneously optimize bond dimensions and assign quality scores to dimension trees with low computational overhead (Section 3.3).

- We implement the proposed ideas as a tool named CADET (CAnonical DimEnsion Trees). Through empirical evaluations, we demonstrate that CADET runs significantly faster, achieves better compression ratios, and scales to larger tensors that baselines cannot handle. Additionally, the discovered topologies can be generalized to unseen data from the same source (Section 5).

## 1.2 Related Work

**Tensor Network Structure Search** TN-SS has been explored in prior work from different aspects. Some studies (Rai et al., 2014; Mickelin & Karaman, 2020; Sedighin et al., 2021; Yin et al., 2022; Ghadiri et al., 2023) focus on optimizing rank assignments for specific tensor network topologies, proposing efficient algorithms to enumerate rank assignments against error constraints. Others focus on topology search with fixed internal ranks (Li & Sun, 2020; Hashemizadeh et al., 2020; Hikihara et al., 2023; Haberstich et al., 2023). Recent work (Li et al., 2022; 2023; Zheng et al., 2024; Zeng et al., 2024; Iacovides et al., 2025; Wang et al., 2026) addresses the complete TN-SS problem by integrating topology and rank search, which often uses sampling-

based methods and verifies if sampled structures satisfy error bounds. Two recent work SVDinsTN (Zheng et al., 2024) and RGTN (Wang et al., 2026) introduce alternatives that encode topology and rank search into a single optimization problem. However, they solve the optimization problem in different ways: SVDinsTN optimizes the objectives with the alternating direction method of multipliers, while RGTN adopts a multi-scale solving strategy to find structures that reach the target error bound. Our approach evolves traditional sampling-based methods by decoupling topology and rank search. We utilize canonical dimension trees to prune redundant candidates and a constraint-based scoring framework to accelerate candidate assessment. As demonstrated in Section 5.2, these strategies allow our method to identify structures with comparable or superior compression performance in significantly less search time than these baselines.

**Low-Rank Tensor Decomposition**   Low-rank tensor decomposition has been studied for many different structures. CANDECOMP/PARAFAC (CP) decomposes a tensor as a sum of rank-one tensors (Kolda & Bader, 2009b; Battaglino et al., 2018; Zhou et al., 2019). Tucker decomposition (Tucker, 1966; De Lath-auwer et al., 2000; Minster et al., 2020; Sun et al., 2020) factorizes a high-order tensor into a core tensor with several low-rank tensors, one for each mode. Tensor train decomposition (Oseledets, 2011; Aksoy et al., 2024) expresses a high-order tensor as a linear multiplication of 3-order tensors. Hierarchical Tucker decomposition (Hackbusch & Kühn, 2009; Grasedyck, 2010; Falcó et al., 2021) generalizes tensor trains and tuckers to arbitrary trees, offering greater flexibility and compression potential. Beyond tree-based structures, several prior work (Espig et al., 2012; Handschuh, 2015; Zhao et al., 2016; Yang et al., 2017; Mickelin & Karaman, 2020) explores cyclic structures including tensor rings and tensor chains. For a more comprehensive overview of tensor decomposition methodologies, we refer the readers to existing surveys (Kolda & Bader, 2009a; Panagakis et al., 2021). In this work, we focus on automatic search of well-performing tree structures to compress input data tensors with a prescribed error bound rather than optimizing ranks or index permutations for fixed structures. Cyclic structures are left as future work.

**Tree Generation**   The systematic generation of tree structures is a foundational problem in enumerative combinatorics, governed by the growth patterns of Catalan and Schröder numbers (Knuth, 1997; Stanley, 2015). Established algorithms for the exhaustive enumeration of these structures typically focus on the Constant Amortized Time (CAT) generation of non-isomorphic unlabeled trees using lexicographical sequences (Aho & Hopcroft, 1974; Beyer & Hedetniemi, 1980; Zaks, 1980; Erdős & Székely, 1989; Sawada, 2006; Kobayashi et al., 2025). In computational biology, these principles are adapted to model hierarchical complexity, such as in phylogenetics, where trees represent evolutionary lineages inferred from genetic data (Semple et al., 2003; Wirtz, 2022; Johnson, 2012). Our work on dimension tree enumeration builds upon these combinatorial foundations but introduces a canonical form specialized for tensor network skeletons. Because tensor networks are inherently unrooted trees, they can be represented by multiple distinct rooted trees; our framework resolves this redundancy by defining a unique representative for each isomorphism class. This symmetry-breaking approach effectively prunes the search space.

## 2   Preliminaries

**Definition 2.1** (Tensor, Tensor Size). Let $d \in \mathbb{N}$ and $n_1, n_2, \ldots, n_d \in \mathbb{N}$. A tensor $\mathcal{X} \in \mathbb{R}^{n_1 \times n_2 \times \cdots \times n_d}$ is a $d$-dimensional array. The $\mu^{th}$ dimension of $\mathcal{X}$ has a name $I_\mu$ with size $n_\mu$ for all $\mu \in \{1, 2, \ldots, d\}$. The size of the tensor $\mathcal{X}$ is defined as $\texttt{size}(\mathcal{X}) = \prod_{\mu=1}^{d} n_\mu$.

**Definition 2.2** (Matricization (Kolda & Bader, 2009b)). For a $d$-dimensional tensor $\mathcal{X} \in \mathbb{R}^{n_1 \times n_2 \times \cdots \times n_d}$ with indices $\{I_1, \ldots, I_d\}$, let $\pi_1, \pi_2, \ldots \pi_d$ be a permutation of $1, 2, \ldots, d$. We partition the dimensions of $\mathcal{X}$ into $\mathcal{I}_s = \{I_{\pi_1}, \ldots, I_{\pi_k}\}$ and $\overline{\mathcal{I}}_s = \{I_{\pi_{k+1}}, \ldots, I_{\pi_d}\}$ such that $\mathcal{I}_s \cap \overline{\mathcal{I}}_s = \emptyset$ and $\mathcal{I}_s \cup \overline{\mathcal{I}}_s = \{I_1, I_2, \ldots, I_d\}$. The $\mathcal{I}_s$-*matricization* of $\mathcal{X}$ is

$$\mathcal{X}^{(\mathcal{I}_s)} = \mathcal{X}(i_{\pi_1}, \ldots, i_{\pi_k}; i_{\pi_{k+1}}, \ldots, i_{\pi_d}) \tag{1}$$

In other words, indices in $\mathcal{I}_s$ enumerate the rows of $\mathcal{X}^{(\mathcal{I}_s)}$, and the remaining indices enumerate the columns.

**Definition 2.3** (Tensor Contraction, Decomposition, and Truncated Decomposition). The contraction between two tensors $\mathcal{A} \in \mathbb{R}^{n_1 \times n_2 \times \cdots \times n_\mu}$ and $\mathcal{B} \in \mathbb{R}^{n_\mu \times n_{\mu+1} \times \cdots \times n_d}$ along the $\mu^{th}$ dimension of $\mathcal{A}$ and first

dimension of $\mathcal{B}$ is an operation represented as $\mathcal{A} \bigotimes_{\mu}^{1} \mathcal{B}$. The resulting tensor $\mathcal{C}$ is computed as

$$\mathcal{C}(i_1, \ldots, i_{\mu-1}, i_{\mu+1}, \ldots, i_d) = \sum_{k=1}^{n_\mu} \mathcal{A}(i_1, \ldots, i_{\mu-1}, k) \times \mathcal{B}(k, i_{\mu+1}, \ldots, i_d) \tag{2}$$

The inverse operation of contraction is decomposition. A tensor $\mathcal{C} \in \mathbb{R}^{n_1 \times n_2 \times n_{\mu-1} \times n_{\mu+1} \times \cdots \times n_d}$ can be decomposed into two smaller tensors $\mathcal{A}$ and $\mathcal{B}$ with respect to a partition of its indices $\{I_i\}_{i=1}^{\mu-1}|\{I_j\}_{j=\mu+1}^{d}$ [1], written as $\mathcal{C} = \mathcal{A} \bigotimes_{\mu}^{1} \mathcal{B}$, where $\mathcal{A} \in \mathbb{R}^{n_1 \times n_2 \times \cdots \times n_\mu}$, $\mathcal{B} \in \mathbb{R}^{n_\mu \times n_{\mu+1} \times \cdots \times n_d}$, and $n_\mu = \min(\prod_{i=1}^{\mu-1} n_i, \prod_{i=\mu+1}^{d} n_i)$. Such decomposition can be done through QR or singular value decomposition (SVD).

A rank-$r$ truncated decomposition of $\mathcal{C}$ is the operation that approximates $\mathcal{C}$ with $\mathcal{A} \bigotimes_{\mu}^{1} \mathcal{B}$ such that the contraction dimension size $n_\mu = r$. This operation is realized through truncated SVD (Hansen, 1987): given a matrix $M$, suppose its SVD result is $M = U\Sigma V$ where $U = [u_1, u_2, \ldots, u_n]$, $V = [v_1, v_2, \ldots, v_n]^T$, and $\Sigma = \text{diag}(\sigma_1, \sigma_2, \ldots, \sigma_n)$ contains singular values in descending order, then its rank $r$ truncation is $\hat{M} = U[:, : r]\Sigma[: r, : r]V[: r]$. This can be easily extended to high-order tensors when reshaping is inserted to make the conversion between matrices and tensors.

**Definition 2.4** (Tensor Network, Tree Tensor Network). A tensor network is an undirected graph $\mathcal{N} = (\mathcal{V}, \mathcal{E})$ where vertices $\mathcal{V}$ are tensors, and edges $\mathcal{E}$ are tuples of two node names and their shared index name (Ye & Lim, 2018). Tensor networks without cycles are called *tree tensor networks* (Falcó et al., 2021). The tensor represented by a graph $\mathcal{N}$ is the contraction of all tensors over shared modes, denoted by $\mathcal{R}_\mathcal{N}$. The size of a tensor network is $\text{size}(\mathcal{N}) = \sum_{\mathcal{X} \in \mathcal{V}} \text{size}(\mathcal{X})$. We call edges with a dangling end *free indices*, and those without dangling ends *contracted indices*. For example, in Fig. 1, the resulting network structure has four nodes, four free indices $I_1, I_2, I_3, I_4$ and three contracted indices $r_1, r_2, r_3$. The represented tensor of this network is

$$\mathcal{R}_\mathcal{N}(i, j, k, l) = \sum_{a=1}^{r_1} \sum_{b=1}^{r_2} \sum_{c=1}^{r_3} \mathcal{X}_1(i, a) \times \mathcal{X}_2(j, b) \times \mathcal{X}_3(a, b, c) \times \mathcal{X}_4(c, k, l) \tag{3}$$

**Definition 2.5** (Tensor Network Structure Search). A TN-SS problem is a tuple $(\mathcal{X}, \varepsilon)$, where $\mathcal{X}$ is the data tensor and $\varepsilon$ is a prescribed error bound. The goal of the TN-SS algorithm is to solve the optimization problem

$$\arg \min_{\mathcal{N}} \ \text{size}(\mathcal{N}) \ \text{s.t.} \ \|\mathcal{R}_\mathcal{N} - \mathcal{X}\|_F \leq \varepsilon \|\mathcal{X}\|_F \tag{4}$$

In other words, the TN-SS problem aims at finding the most compressed tensor network within a given error bound. In this work, we target arbitrary tree structures, excluding structures with cycles.

## 3 TN-SS via Canonical Dimension Tree Enumeration

This section details the proposed algorithm CADET. We first present the high level pipeline of the algorithm. Followed by that, we formalize the search space through the lens of dimension trees. Finally, we introduce the dimension tree scoring method, which utilizes precomputed singular values to estimate approximation errors without incurring the cost of iterative SVDs.

### 3.1 The High-Level Algorithm

Algorithm 1 outlines the primary stages of the CADET framework. The workflow adopts a paradigm of candidate enumeration followed by systematic assessment, utilizing a priority queue $\mathcal{C}$ to maintain the highest-quality structures identified during the search (Line 3). The algorithm iteratively traverses the search space by generating candidate dimension trees (Lines 4–6). Within this loop, the quality of each candidate $T_\mathcal{I}$ is evaluated by calculating a cost $c$ (Line 5), which determines the priority of the structure within $\mathcal{C}$ (Line 6). Once the top-$k_\theta$ candidate structures are established, full tensor decompositions are performed to compress the input data tensor $\mathcal{X}$ into these specific topologies, ultimately returning the structure that yields the minimum tensor network size (Lines 7–10).

---

[1] We use $\{X_i\}_{i=a}^{b}$ to represent $\{X_a, X_{a+1}, \ldots, X_b\}$

---

**Algorithm 1** Proposed tensor network structure search algorithm

---

**Input** Data tensor $\mathcal{X}$, error bound $\varepsilon$, number of selected candidate structures $k_\theta$, and maximum number of nodes in the dimension tree $S_\theta$

**Output** A tensor network $\mathcal{N}$ such that $\texttt{size}(\mathcal{N}) \leq \texttt{size}(\mathcal{X})$, and $\|\mathcal{R}_\mathcal{N} - \mathcal{X}\|_F \leq \varepsilon \|\mathcal{X}\|_F$

1: **function** CADET($\mathcal{X}, \varepsilon, k_\theta, S_\theta$)
2:     $\Omega \leftarrow$ PREPROCESS($\mathcal{X}, \varepsilon$)     ▷ Precompute singular values for all tensor matricizations (Section 3.3.1)
3:     $\mathcal{C} \leftarrow \emptyset$
4:     **for** $T_\mathcal{I} \in$ ENUMDIMTREES(INDICES($\mathcal{X}$), $S_\theta$) **do**        ▷ Enumerate candidate dim trees (Section 3.2)
5:         $c, \rho \leftarrow$ GETCOST($\Omega, \mathcal{X}, \varepsilon, T_\mathcal{I}$)                ▷ Compute cost via constraint solving (Section 3.3.2)
6:         add $(T_\mathcal{I}, c, \rho)$ to $\mathcal{C}$ and remove trees with large costs to maintain $|\mathcal{C}| \leq k_\theta$
7:     $\mathcal{N}_{\min} \leftarrow (\{\mathcal{X}\}, \emptyset)$
8:     **for** $(T_\mathcal{I}, c, \rho) \in \mathcal{C}$ **do**
9:         $\mathcal{N}_{\min} \leftarrow \min(\mathcal{N}_{\min}, \text{DECOMPOSE}(\mathcal{X}, T_\mathcal{I}, \varepsilon, \rho))$   ▷ Decompose $\mathcal{X}$ into $T_\mathcal{I}$ with ranks $\rho$ (Section 3.4)
10:     **return** $\mathcal{N}_{\min}$

---

There are two technical insights in this algorithm design. First, the search space is restricted to a specific subset of dimension trees to effectively exclude suboptimal and duplicate network skeletons from the enumeration process (Section 3.2). Second, CADET eliminates the requirement for tensor decompositions during the screening of candidates. By executing a one-time preprocessing of the data tensor to construct a metadata map $\Omega$, the algorithm stores the singular values for all valid index partitions, or tensor matricizations (Line 2). Leveraging the property that singular values associated with a specific index partition are monotonically non-increasing following tensor truncation, the framework utilizes these precomputed singular values to implement a computationally efficient proxy for candidate cost estimation (Section 3.3).

### 3.2 Network Topology Exploration via Dimension Tree Enumeration

In this section, we establish a systematic framework for navigating the space of tensor network topologies. We first introduce *generalized dimension trees* (GDTs) as a formal representation of network topologies. By relaxing the exhaustive partitioning requirements of traditional dimension trees (Grasedyck, 2010; Falcó et al., 2021), GDTs can represent a significantly broader class of networks that allow multiple free indices on both leaf and non-leaf nodes.

To ensure search efficiency, we incorporate pruning rules within this representation to eliminate suboptimal topologies early in the process. Furthermore, since a single network topology can be mapped to multiple valid GDTs, we define a *canonical dimension tree* (CDT) that establishes a one-to-one mapping between the tree structure and the underlying topology, effectively removing redundancy during the enumeration phase.

**Definition 3.1** (Generalized Dimension Trees)**.** Let $\mathcal{I} = \{I_1, \ldots, I_d\}$ be the set of free indices associated with a $d$-dimensional tree tensor network $\mathcal{N}$. A tree $T_\mathcal{I}$ is a generalized dimension tree of $\mathcal{I}$ if

(a) The root node is the full set $\mathcal{I}$; every other node $v$ corresponds to a non-trivial subset of $\mathcal{I}$;

(b) For any node $v$, its children $\{c_1, c_2, \ldots, c_k\}$ constitute non-trivial, mutually disjoint subsets of $v$, such that $c_i \subset v, c_i \neq \emptyset$ for all $i$ and $c_i \cap c_j = \emptyset$ for all $i \neq j$;

(c) For any pair of children $u, v$ of the root, $u \cup v \neq \mathcal{I}$.

Nodes with no children are called leaf nodes. We use $\texttt{children}(T_\mathcal{I}, \mathcal{I}_s)$ to represent children nodes of $\mathcal{I}_s \in T_\mathcal{I}$.

The main differences between GDTs and traditional dimension trees are requirements (b), and (c). Requirement (b) relaxes the exhaustive partitioning of indices among children found in traditional dimension trees. Instead, it allows a node to contain indices that do not appear in any of its children; these indices are thus designated as free indices of that specific node. Requirement (c) excludes structures with suboptimal nodes.

Figure 2 illustrates the versatility of GDTs in representing diverse tensor network topologies. Unlike conventional dimension trees where the union of children's indices must exhaustively cover the parent's set, this

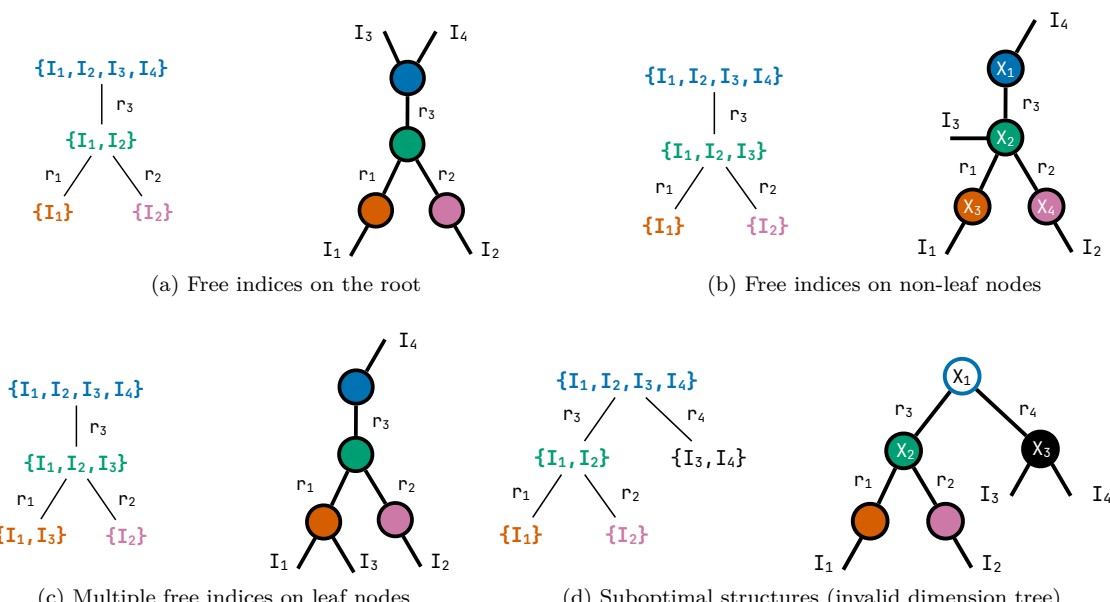

Figure 2: Examples of generalized dimension trees and their corresponding tensor network skeletons. Unfilled nodes represent structures that lead to sub-optimality, which can be merged with adjacent neighbors to reduce the total network size. Node labels in (b) and (d) are provided for ease of reference in the text.

generalized framework accommodates indices at multiple hierarchical levels. As shown in Fig. 2a and Fig. 2b, free indices can be associated directly with the root and other non-leaf nodes, respectively. Furthermore, Fig. 2c demonstrates that leaf nodes may support multiple free indices simultaneously. This representation allows for the identification of suboptimal configurations; for instance, the dimension tree in Fig. 2d is invalid because it contains two nodes $\{I_1, I_2\}$ and $\{I_3, I_4\}$ that sums up to the total index set, violating the requirement (c) of dimension tree's definition. While this structure resembles a traditional Hierarchical Tucker (HT) format (Grasedyck, 2010), the root node $\mathcal{X}_1$ serves no purpose and only adds unnecessary parameters. Specifically, without merging, the total size of nodes $\mathcal{X}_1$, $\mathcal{X}_2$, and $\mathcal{X}_3$ is $r_3 r_4 + r_1 r_2 r_3 + r_4 n_3 n_4$. The uncolored node $\mathcal{X}_1$ can be merged with adjacent neighbors to reduce the total network size: if $r_3 < r_4$, $\mathcal{X}_1$ should be merged to the right neighbor $\mathcal{X}_3$, reducing the total size into $r_1 r_2 r_3 + r_3 n_3 n_4$; if $r_3 > r_4$, $\mathcal{X}_1$ should be merged to the right neighbor $\mathcal{X}_2$, reducing the total size into $r_1 r_2 r_4 + r_4 n_3 n_4$. Both reduced sizes are strictly lower than the unmerged sizes. To summarize, this GDT mapping ensures that the search space is both inclusive of necessary topologies and pruned of inherently inefficient skeletons.

**Definition 3.2** (Generalized Dimension Tree Size). Given a generalized dimension tree $T_{\mathcal{I}}$ for free indices $\mathcal{I} = \{I_1, \ldots, I_d\}$ with $n+1$ nodes $\mathcal{I}_0 = \mathcal{I}$ and $\mathcal{I}_1, \cdots, \mathcal{I}_n \subset \mathcal{I}$, and a rank assignment $\rho = \{r_i\}_{i=1}^n$ where $r_i$ is the rank size for the edge between nodes $\mathcal{I}_i$ and its parent, and $r_0 = 1$, the size of the dimension tree $T_{\mathcal{I}}$ is recursively defined as

$$\texttt{size}\left(T_{\mathcal{I}} \mid \{r_i\}_{i=1}^n\right) = \sum_{i=0}^n \left( r_i \times \prod_{I \in \mathcal{I}_i \setminus \cup_c \mathcal{I}_c} \texttt{size}\left(I\right) \times \prod_c r_c + \sum_c \texttt{size}\left(T_{\mathcal{I}_c} \mid \{r_i\}_{i=1}^n\right) \right) \quad (5)$$

where $\mathcal{I}_c$ are the children of the node $\mathcal{I}_i$, and $T_{\mathcal{I}_c}$ are the subtrees rooted at each child node.

Intuitively, the size of a dimension tree sums up the size of all nodes inside it. As an example, consider the dimension tree in Fig. 2b with rank assignments $r_1, r_2, r_3$ corresponding to the edges between $(\mathcal{X}_2, \mathcal{X}_3)$,

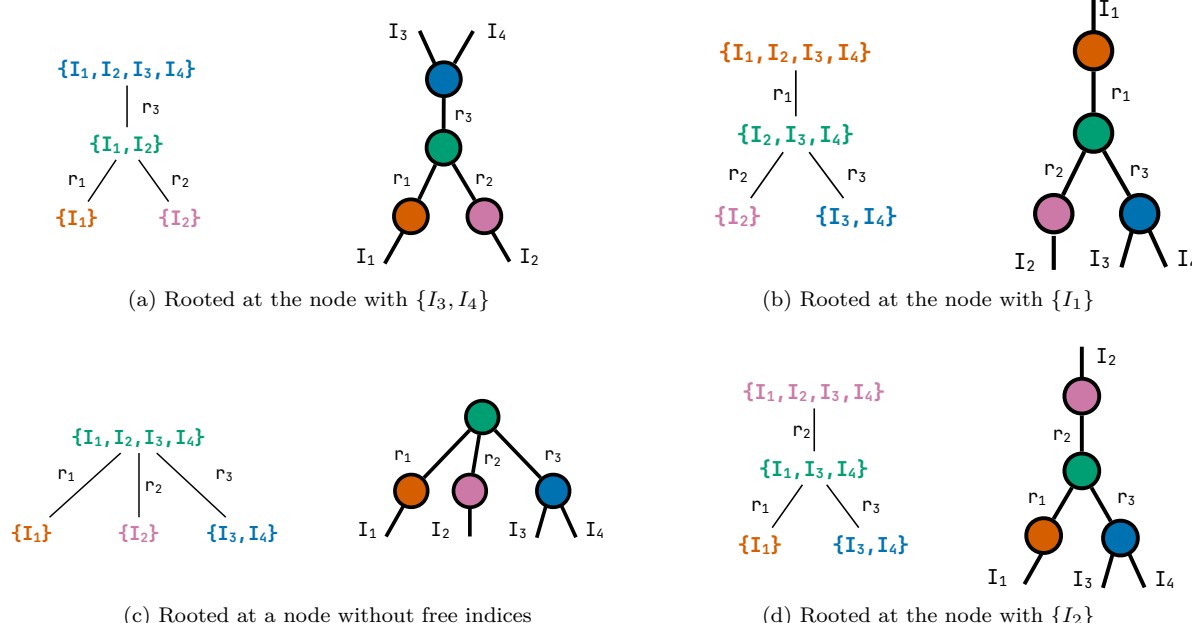

Figure 3: Generalized dimension trees for the same tensor network skeleton at different roots. Fig. 3a is in canonical form whereas the other three are not.

$(\mathcal{X}_2, \mathcal{X}_4)$, and $(\mathcal{X}_1, \mathcal{X}_2)$ respectively. Then the size of this dimension tree is

$$
\begin{aligned}
\texttt{size}\left(T_{I_1,I_2,I_3,I_4} \mid \{r_1, r_2, r_3\}\right) = &\ \texttt{size}\left(I_4\right) \times r_3 && \text{size of node } \mathcal{X}_1 \\
&+ r_3 \times \texttt{size}\left(I_3\right) \times r_1 \times r_2 && \text{size of node } \mathcal{X}_2 \\
&+ r_1 \times \texttt{size}\left(I_1\right) && \text{size of node } \mathcal{X}_3 \\
&+ r_2 \times \texttt{size}\left(I_2\right) && \text{size of node } \mathcal{X}_4
\end{aligned}
$$

However, GDTs are not unique to a given tensor network. As illustrated in Fig. 3, a single network can yield multiple distinct dimension trees depending on the node selected as the root. This non-uniqueness introduces a significant challenge for network skeleton enumeration: the search algorithm must effectively prune redundant exploration paths to avoid the computational cost of processing equivalent structures. To eliminate this redundancy, we define a canonical dimension tree (CDT) for each tensor network and restrict the enumeration process exclusively to these canonical forms.

**Definition 3.3** (Order on Index Subsets). Let $\mathcal{I} = \{I_i\}_{i=1}^d$ be a set of indices with a lexicographical order such that $I_1 < I_2 < \cdots < I_d$. For any two subsets $\mathcal{I}_a, \mathcal{I}_b \subseteq \mathcal{I}$, we define a total order $\mathcal{I}_a < \mathcal{I}_b$ according to the following criteria:

  (a) Cardinality: $\mathcal{I}_a < \mathcal{I}_b$ if $|\mathcal{I}_a| < |\mathcal{I}_b|$

  (b) Lexicographical precedence: If $|\mathcal{I}_a| = |\mathcal{I}_b| = k$, and their respective sorted elements are $(a_1, \ldots, a_k)$ and $(b_1, \ldots, b_k)$, then $\mathcal{I}_a < \mathcal{I}_b$ if there exists $1 \leq j \leq k$ such that $a_j < b_j$ and $a_i = b_i$ for all $i < j$.

Under this ordering, the subsets follow the sequence $\{I_1\} < \{I_2\} < \{I_1, I_2\} < \{I_1, I_4\} < \{I_2, I_3\}$. This ordering allows us to uniquely identify a representative structure, which we define as follows:

**Definition 3.4** (Canonical Dimension Trees). Let $\mathcal{I}$ be the set of $d$ free indices of a tree tensor network $\mathcal{N}$. A generalized dimension tree $T_\mathcal{I}$ is considered a canonical dimension tree of $\mathcal{N}$ if every node $\mathcal{I}_s \in T_\mathcal{I}$ satisfies either $\mathcal{I}_s = \mathcal{I}$ or $\mathcal{I}_s < \mathcal{I} \setminus \mathcal{I}_s$.

---

**Algorithm 2** Discover all possible canonical dimension trees for a given data tensor.

---

**Input** A set of free indices $\mathcal{I}$, and maximum number of nodes in the resulting dimension tree $S_\theta$
**Output** A set of canonical dimension trees $T_\mathcal{I}$

 1: **function** ENUMDIMTREES($\mathcal{I}, S_\theta$)
 2:      Initialize $T_\mathcal{I}$ with $\mathcal{I}$ being the root
 3:      **yield** from GROWTREE($T_\mathcal{I}, S_\theta$)

 4: **function** GROWTREE($T_\mathcal{I}, S_\theta$)
 5:      **if** size $(T_\mathcal{I}) \geq S_\theta$ **then return**
 6:      **for** $\mathcal{I}_s \in T_\mathcal{I}$ **do**                                 ▷ Enumerate all possible expansions
 7:         $\mathcal{I}_r \leftarrow \mathcal{I}_s \setminus \cup_{c \in \texttt{children}(T_\mathcal{I}, \mathcal{I}_s)} c$          ▷ Get remaining indices on the current node
 8:         **for** $\mathcal{I}_i \subset \mathcal{I}_r$ s.t. $\mathcal{I}_i \neq \emptyset$ **do**            ▷ Separate a subset from remaining indices
 9:            **if** $\exists \mathcal{I}_j \in T_\mathcal{I}. \; \mathcal{I}_j \cup \mathcal{I}_i = \mathcal{I}$ **then continue**    ▷ Well-formedness of generalized dimension trees
10:            **if** $\mathcal{I} \setminus \mathcal{I}_i < \mathcal{I}_i$ **then continue**              ▷ Canonical form of dimension trees
11:            **if** $\exists \mathcal{I}_j \in T_\mathcal{I}. \; \mathcal{I}_j < \mathcal{I}_i$ **then continue**     ▷ Enforce the expansion order to avoid duplicates
12:            Add $\mathcal{I}_i$ to $T_\mathcal{I}$ as a child of $\mathcal{I}_s$ and **yield** $T_\mathcal{I}$
13:            GROWTREE($T_\mathcal{I}, S_\theta$)

---

Among the four GDTs illustrated in Fig. 3, only Fig. 3a is in canonical form. The other trees fail this criterion due to specific node violations: the nodes $\{I_2, I_3, I_4\}$ in Fig. 3b, $\{I_3, I_4\}$ in Fig. 3c, and $\{I_1, I_3, I_4\}$ in Fig. 3d all violate the canonical condition because their respective complement sets are smaller under the order in Definition 3.3.

### 3.2.1 Dimension Tree Enumeration (Procedure EnumDimTrees)

The generation of CDTs is formalized in Algorithm 2. The algorithm takes a set of free indices $\mathcal{I}$ and a parameter $S_\theta$ to control the size of output dimension trees. At a high level, the procedure initializes a tree with a single root node $\mathcal{I}$ (Line 2) in accordance with dimension tree's definition rule (a), and subsequently expands the structure by recursively inserting child nodes (Line 3).

The core logic resides within the GROWTREE function, which employs an iterator-style approach to yield valid canonical trees. GROWTREE first verifies the current tree size against the limit $S_\theta$ (Line 5). If capacity remains, the algorithm iterates through all existing nodes in the tree (Line 6) to identify potential expansion points. For a selected node, the algorithm identifies the set of indices $\mathcal{I}_r$ not yet partitioned (Line 7) and evaluates all possible non-empty subsets $\mathcal{I}_i \subset \mathcal{I}_r$ to create new child nodes (Line 8). To ensure efficiency and uniqueness, the algorithm applies three critical pruning filters:

(1) **Well-Formedness of Dimension Trees:** A new node $\mathcal{I}_i$ is discarded if, when combined with an existing sibling node $\mathcal{I}_j$, it fully partitions the index set $\mathcal{I}$. This prevents the formation of invalid trees that violate the rule (c) in the definition of dimension trees (Line 9).

(2) **Canonical Form of Dimension Trees:** The algorithm enforces that the enumerated trees are canonical. If the complement set $\mathcal{I} \setminus \mathcal{I}_i$ is smaller than $\mathcal{I}$ according to our defined set order, the node is skipped to ensure only the canonical form is explored (Line 10).

(3) **Insertion Order:** To prevent isomorphic trees arising from different insertion sequences (e.g., adding $\{I_1\}$ then $\{I_2\}$ versus vice-versa), we enforce a strict descending order for node introduction (Line 11). A new node is only added if it is smaller than previously added nodes in the current tree.

Once a candidate node $\mathcal{I}_i$ passes these checks, it is integrated into the tree and yielded as a valid canonical structure (Line 12). The process then continues recursively to explore deeper expansions (Line 13).

Figure 4 illustrates this enumeration process for a four-dimensional tensor. Starting from the single node tree $T_0$, the algorithm introduces child nodes such as $\{I_1, I_2\}$ (step ①), $\{I_4\}$ (step ⑦), etc. to expand the

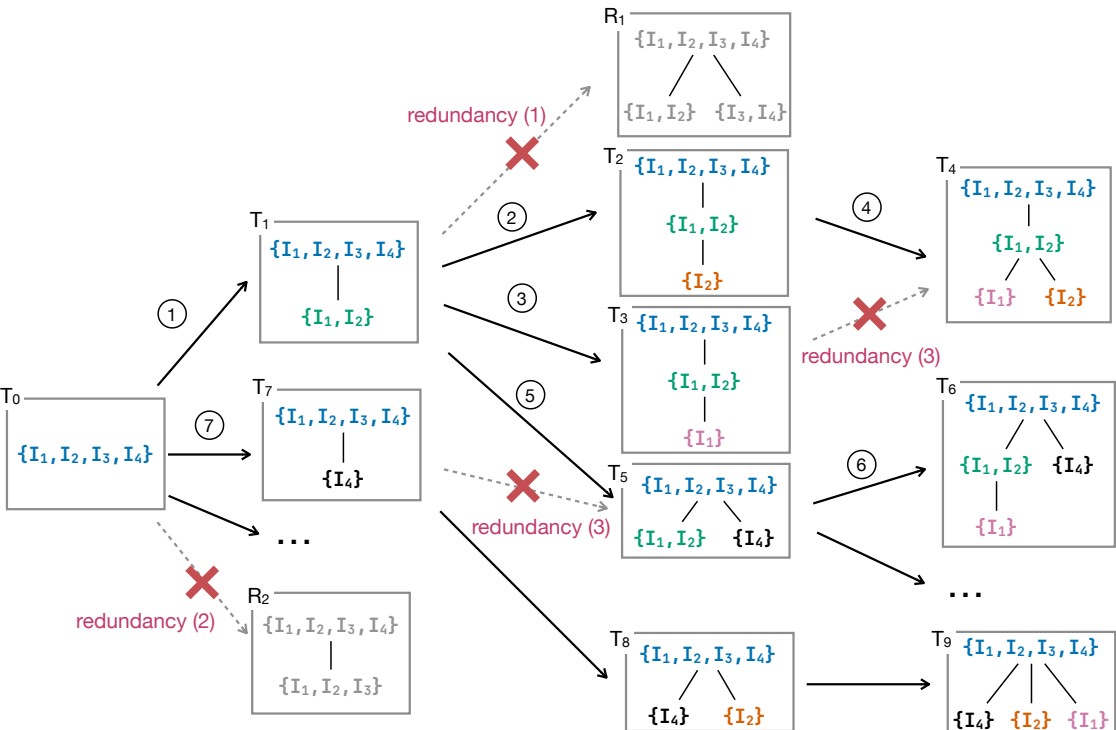

Figure 4: Enumeration of canonical dimension trees. Gray dashed edges and trees are redundant redundant enumeration paths and trees respectively, which are excluded from the search process. Three kinds of redundancy are pruned: (1) invalid dimension trees $R_1$ where the two sibling nodes $\{I_1, I_2\}$ and $\{I_3, I_4\}$ form a full partition of the indices; (2) non-canonical dimension trees $R_2$ where the node $\{I_1, I_2, I_3\}$ has a shorter complement node $\{I_4\}$, and therefore $\{I_1, I_2, I_3\}$ should never appear in a canonical dimension tree; (3) duplicate canonical dimension trees resulted from multiple enumeration paths ($T_3$ to $T_4$, $T_7$ to $T_5$). Redundancy (1) and (2) are ruled out by the definition of canonical dimension trees, while redundancy (3) is excluded by enforcing the order when adding nodes.

structure. Among them, the transition from $T_0$ to $R_2$ is excluded because $R_2$ contains the node $\{I_1, I_2, I_3\}$, violating the canonical rule of CDTs as its complement set $\{I_4\}$ is smaller in our defined order. Focusing on the valid path in step ①, the algorithm introduces the child $\{I_1, I_2\}$. From this state, the tree can grow by attaching $\{I_2\}$ (step ②) or $\{I_1\}$ (step ③) to the leaf $\{I_1, I_2\}$, or by adding a sibling node $\{I_4\}$ to the root (step ⑤). Notably, the algorithm prohibits attaching $\{I_3, I_4\}$ to the root (tree $R_1$) as it would violate the well-formedness rule of GDTs. Furthermore, while steps ② and ③ could eventually converge on the same structure $T_4$, only the path through step ④ is permitted; the transition from $T_3$ to $T_4$ is pruned by the insertion order constraint because $\{I_1\} < \{I_2\}$.

### 3.3 Fast Scoring Via Constraint Solving

The CDTs generated via enumeration define an expansive search space of tensor network skeletons. While these structures establish the node connectivity (topology), the associated edge weights, specifically the rank assignments for each bond, remain undetermined. A naïve approach would involve executing a full tensor decomposition for every candidate tree $T_{\mathcal{I}}$ to evaluate its performance. However, such a strategy is computationally prohibitive for large data tensors, as the cost of decomposition scales poorly regardless of the specific algorithm employed.

To bypass this bottleneck, we propose a constraint-solving technique designed to rapidly screen candidate structures and figure out the near-optimal rank assignment at the same time. This method identifies rank configurations with the highest compression potential, significantly reducing the number of full-scale decom-

---

**Algorithm 3** Precompute singular values for all matricizations of a given data tensor.

---

**Input** A set of free indices $\mathcal{I}$
**Output** Mapping $\Omega$ from index subsets to singular values
 1: **function** PREPROCESS($\mathcal{X}$)
 2:      $\mathcal{I} \leftarrow$ INDICES($\mathcal{X}$)
 3:      $\Omega \leftarrow \emptyset$
 4:      **for** $s = 1, 2, \ldots, \lfloor \frac{\texttt{len}(\mathcal{I})}{2} \rfloor$ **do**                        ▷ Enumerate all possible index subsets
 5:          **for** $\mathcal{I}_s \in$ SUBSETSOF($\mathcal{I}, s$) **do**
 6:              $U; \Sigma; V \leftarrow$ SVD($\mathcal{X}^{(\mathcal{I}_s)}$)
 7:              $\Omega[\mathcal{I}_s] \leftarrow \texttt{diag}(\Sigma)$             ▷ Store the singular values for the unfolding at $\mathcal{I}_s$
 8:      **return** $\Omega$

---

positions required. The fundamental insight of this approach is that the singular value spectra of the original data tensor $\mathcal{X}$ can serve as a rigorous upper bound (over-approximation) for the singular values encountered during the actual hierarchical decomposition process.

In the following sections, we detail the mechanics of this scoring framework. We first describe the one-time singular value precomputation phase (Line 2 of Algorithm 1), which populates the metadata map $\Omega$. We then demonstrate how these precomputed singular values are utilized in a fast cost computation routine (Line 5) to evaluate candidate skeletons without the need for intermediate tensor factorizations.

### 3.3.1 Pre-computation of Singular Values (Procedure Preprocess)

At the initialization of CADET (Line 2), we construct a metadata map $\Omega$ to store the singular values for all possible matricizations of the input data tensor. As detailed in Algorithm 3, the tensor $\mathcal{X}$ is unfolded along every subset of its free indices $\mathcal{I}$ (Line 4). For each matricization $\mathcal{X}^{(\mathcal{I}_s)}$, we perform SVD to extract the associated singular values (Line 6). These information are indexed by their corresponding index sets $\mathcal{I}_s$ and archived in $\Omega$ (Line 7). While this preprocessing is the most computationally expensive phase of the algorithm with a complexity that grows exponentially with the number of dimensions, it is a one-time cost that serves as a high-speed look-up table for the assessment phase. Since the algorithm bypasses online decompositions during the evaluation of thousands of candidate skeletons, it remains highly efficient. For very high-dimensional tensors, this costly preprocessing step can be embedded within a hierarchical search framework to reduce the high computation complexity (Guo et al., 2026).

### 3.3.2 Constraint-Based Rank Optimization and Assessment (Procedure GetCost)

While CADET can exhaustively enumerate a vast library of dimension trees, each representing a distinct tensor network skeleton, it is computationally prohibitive to perform explicit tensor decompositions to identify the best rank assignment for every candidate. Even with fixed ranks, the cost of iterative factorizations remains the primary bottleneck in the search process (Section 5.3). To overcome this, we treat rank assignment not as a search variable, but as a constrained optimization problem solved analytically for each dimension tree.

Given a data tensor $\mathcal{X}$, an error bound $\varepsilon$, and a dimension tree $T_{\mathcal{I}}$ with $n + 1$ nodes where $\mathcal{I}_0 = \mathcal{I}$ and $\mathcal{I}_1, \mathcal{I}_2, \ldots, \mathcal{I}_n \subset \mathcal{I}$, we formulate the task of GETCOST as finding the rank assignment $\rho = \{r_s\}_{s=1}^n$ that minimizes the dimension tree size while satisfying the error budget. Leveraging the precomputed singular value in $\Omega$, we model this as the following integer programming problem:

$$\min_{r_1, r_2, \ldots, r_n} \texttt{size}\left(T_{\mathcal{I}} \mid \{r_s\}_{s=1}^n\right) \quad \text{s.t.} \quad \sum_{s=1}^n \sum_{i > r_s} \Omega^2[\mathcal{I}_s, i] \leq (\varepsilon \|\mathcal{X}\|_F)^2 \tag{6}$$

In this formulation, each $r_s$ is an integer variable for the edge rank between node $\mathcal{I}_s$ and its parent, and $\Omega[\mathcal{I}_s, i]$ represents the $i^{th}$ largest singular value for the $\mathcal{I}_s$ unfolding of data tensor $\mathcal{X}$. The constraint ensures that the cumulative truncation error, which is equal to the sum of discarded singular values across all $n$ edges, remains within the threshold $\varepsilon$. Once solved, the resulting rank assignment $\rho$ is recorded and the corresponding dimension tree size serve as the cost metric used to rank the trees.

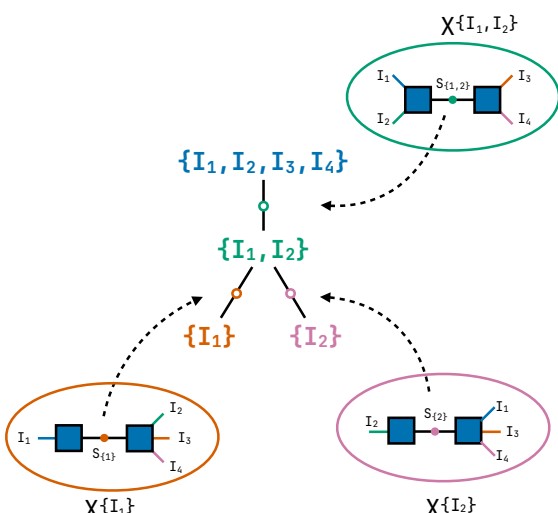

Figure 5: Approximation of singular values during decomposition using those of the input data tensor. Big circles denote the pre-computed SVD results over different matricizations of the input tensor. Solid small dots represent singular values for the data tensor, and unfilled small circles represent the location of what singular values we are approximating.

As illustrated in Fig. 5, this approach approximate hierarchical tensor decompositions described in a dimension tree with a sequence of independent tensor decompositions applied to the original data tensor. For a dimension tree defined by the nested index sets $\{I_1, I_2, I_3, I_4\}$, $\{I_1, I_2\}$, $\{I_2\}$, and $\{I_1\}$, we approximate the singular values at each edge using the precomputed $\Omega[\{I_1, I_2\}]$, $\Omega[\{I_2\}]$, and $\Omega[\{I_1\}]$. This methodology is theorectically supported by the following property:

**Theorem 3.5** (Singular Value Upper Bound). *Let $\mathfrak{X} \in \mathbb{R}^{n_1 \times \cdots \times n_d}$ be a d-dimensional tensor, compressing $\mathfrak{X}$ into a structure described by a dimension tree $T_{\mathcal{I}}$ of $n+1$ nodes $\mathcal{I}, \mathcal{I}_1, \mathcal{I}_2, \ldots, \mathcal{I}_n$ produces the structure $\mathcal{N}$, then for every $1 \leq i, s \leq n$, we have $\sigma_j\left(\mathcal{R}_{\mathcal{N}_i}^{(\mathcal{I}_s)}\right) \leq \sigma_j\left(\mathfrak{X}^{(\mathcal{I}_s)}\right)$ where $\sigma_j(A)$ is the $j^{th}$ largest singular value of the matrix A, and $\mathcal{N}_i$ is the network obtained after the first i tensor decompositions specified by $T_{\mathcal{I}}$.*

*Proof.* By the definition of dimension trees, for every pair $1 \leq s < t \leq n$, there could only be three relations between $\mathcal{I}_s$ and $\mathcal{I}_t$: $\mathcal{I}_s \subset \mathcal{I}_t$, $\mathcal{I}_t \subset \mathcal{I}_s$, or $\mathcal{I}_s \cap \mathcal{I}_t = \emptyset$.

Suppose the network obtained after the $k^{th}$ tensor decomposition is denoted as $\mathcal{N}_k$. The network obtained after performing the tensor decomposition on $\mathcal{N}_k$ along index set $\mathcal{I}_k$ is $\mathcal{N}_{k+1}$. Performing the split defined above is equivalent to performing a truncated SVD on $\mathcal{R}_{\mathcal{N}_k}^{(\mathcal{I}_k)}$. Formally, we can say that if $\mathcal{R}_{\mathcal{N}_k}^{(\mathcal{I}_k)} = U\Sigma V$, then $\mathcal{R}_{\mathcal{N}_{k+1}}^{(\mathcal{I}_k)} = \widetilde{U}\widetilde{\Sigma}\widetilde{V}$, where $\widetilde{U}$, $\widetilde{\Sigma}$, and $\widetilde{V}$ are truncated matrices of $U$, $\Sigma$, and $V$. Consequently, using Theorem A.5, we have that, for $\mathcal{I}_t \subset \{I_1, \ldots, I_d\}$ such that $\mathcal{I}_t \subseteq \mathcal{I}_s, \mathcal{I}_s \subseteq \mathcal{I}_t$, or $\mathcal{I}_s \cap \mathcal{I}_t = \emptyset$, $\sigma_i\left(\mathcal{R}_{\mathcal{N}_{k+1}}\right) \leq \sigma_i\left(\mathcal{R}_{\mathcal{N}_k}\right)$ for all possible $i$ and $k$.

From the above result, we can conclude that $\sigma_i\left(\mathcal{R}_{\mathcal{N}_k}^{(\mathcal{I}_t)}\right) \leq \sigma_i\left(\mathcal{R}_{\mathcal{N}_0}^{(\mathcal{I}_t)}\right) = \sigma_i\left(\mathfrak{X}^{(\mathcal{I}_t)}\right)$ for all valid choices of $\mathcal{I}_t$ and all possible values of $i$ and $k$. $\qquad\square$

Theorem 3.5 allows us to use the singular values of the original data as a rigorous over-approximation for any candidate CDT. While this may lead to a conservative estimate of the truncation error, we find that this upper bound is an exceptionally effective proxy for ranking. By solving for this bound, CADET identifies high-performance CDTs without the prohibitive overhead of live tensor decompositions during search.

### 3.4 Tensor Decomposition From Dimension Trees (Procedure Decompose)

The final step of CADET is the physical transformation of the data tensor into the tensor network structures dictated by the highest-ranked dimension trees (Lines 7–10 of Algorithm 1). Unlike the assessment phase

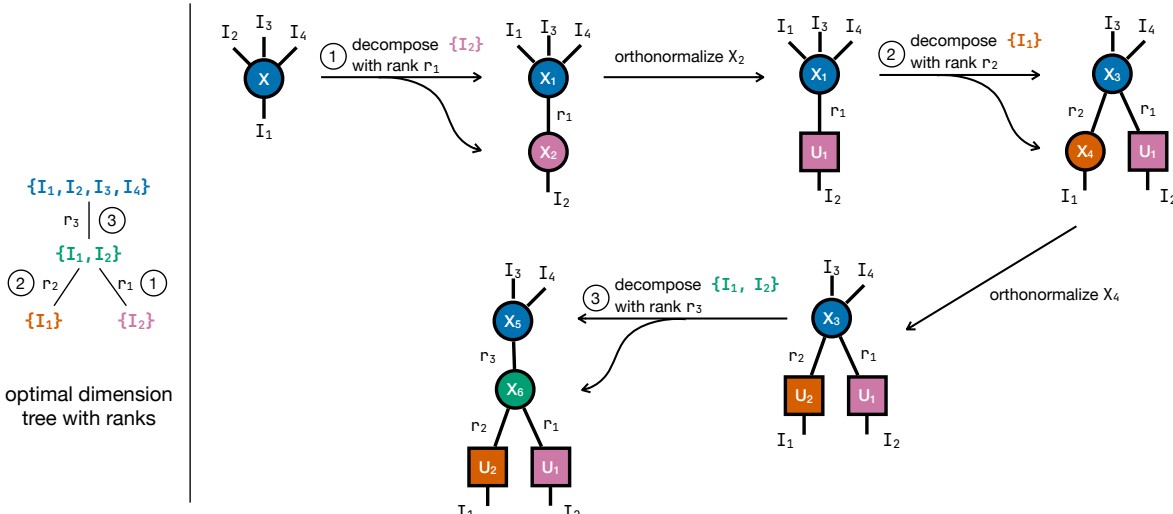

Figure 6: Decompose a data tensor into the format described by a generalized dimension tree. (left) A generalized dimension tree with rank assignments. (right) Step-by-step transformation from a data tensor to the tensor network represented by the dimension tree on the left. The tensor decomposition and network orthonormalization interleaves to ensure truncation error correctly propagates from individual nodes to the entire network. Square nodes stand for orthogonal nodes.

that operates on $\Omega$, the DECOMPOSE procedure executes a sequence of tensor decompositions, using the determined rank assignments $\rho$ as the bond dimensions.

The hierarchical structure of the generalized dimension tree $T_{\mathcal{I}}$ serves as a blueprint for this process. Each non-root node in $T_{\mathcal{I}}$ prescribes a specific bipartition that incrementally shapes the network topology. As illustrated in Fig. 6, the process begins at a leaf representing an index set $\{I_2\}$, and isolates these indices via a $r_1$ truncated SVD along the bipartition $\{I_2\}|\mathcal{I} \setminus \{I_2\}$, splitting the data tensor into two factors (step ①).

When moving to the leaf $\{I_1\}$, the algorithm identifies $\mathcal{X}_1$ as the target node. To ensure the subsequent steps correctly compute the singular values, the algorithm must shift the orthogonality center between splits. Standard tensor network theory dictates that a local SVD only yields the true, global singular values of a bipartition if the operating node concentrates the system's full norm (White, 1992; 1993; Evenbly, 2022). Therefore, the previously created factor $\mathcal{X}_2$ is orthonormalized into the isometry $\mathcal{U}_1$, pushing the global norm into $\mathcal{X}_1$. Then, the second decomposition performed to separate index $I_1$ (step ②).

For the non-leaf node $\{I_1, I_2\}$, the algorithm identifies the node in the current network that serves as the lowest common ancestor to both the $I_1$ and $I_2$ branches. In our example, this common ancestor is $\mathcal{X}_3$. After centering the orthogonality at $\mathcal{X}_3$ by orthonormalizing surrounding tensors, a further decomposition splits it into $\mathcal{X}_5$ and $\mathcal{X}_6$ to achieve the desired index partition (step ③).

Following the completion of the hierarchical decomposition, the resulting network approximates the original data $\mathcal{X}$ within the relative error bound $\varepsilon$. However, as established by Theorem 3.5, the singular values used during the constraint-solving phase are rigorous over-approximations derived from the original tensor. This implies that the solved ranks $\rho$ may be more conservative than strictly necessary (the practical tightness and empirical accuracy of this approximation are validated by the close alignment demonstrated in Table 3). To recover these potential gains, we apply a rounding procedure, an extension of the TT-rounding algorithm (Oseledets, 2011) adapted for tree structures to exhaust the remaining error budget. Notably, we employ the solved ranks $\rho$ in the final decomposition rather than adopting an even error distribution across decomposition stages as common in TT or HT decomposition. In practice, we observe that this strategy yields improved compression ratios by utilizing the specific rank assignment identified by the solver.

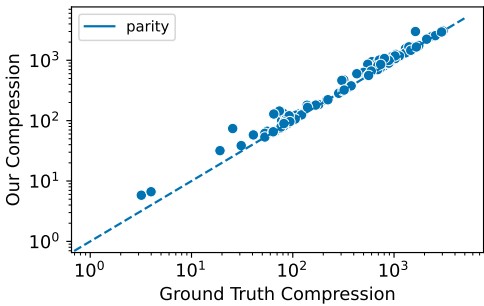

Figure 7: Comparison of compression ratios for random generated data. Ground truth compression is the compression ratio of the structure used to generate synthetic data.

## 4 Complexity Analysis

Given a tensor of $d$ dimensions and each dimension has size $n$, the preprocessing phase runs the SVD for all possible index partitions, which takes time $\mathcal{O}\left(n^{1.5d}2^d\right)$. Then, the algorithm enumerates all dimension trees with at most $S_\theta$ nodes. The total number of dimension trees is $\mathcal{O}\left(2^{S_\theta d}\right)$. For each dimension tree, a constraint solving is run to compute the rank assignment, each taking time $\mathcal{O}\left(S_\theta^{2.5}2^{S_\theta}\right)$. Lastly, we compress the data tensor for the best dimension tree, which involves at most $S_\theta$ truncated SVDs. Therefore, the total complexity is $\mathcal{O}\left(n^{1.5d}(S_\theta + 2^d) + S_\theta^{2.5}2^{S_\theta+S_\theta d}\right)$. This asymptotic complexity analysis demonstrates that preprocessing time scales exponentially with tensor order, a trend clearly reflected in our empirical benchmarks in Table 1. Therefore, we bound our evaluation of CADET to data tensors of order $d \leq 6$ and search spaces of $S_\theta \leq 6$ nodes. Within this practical regime, the algorithm consistently delivers strong, highly competitive compression ratios. Extending this framework to support higher-dimensional tensors efficiently remains an active direction for future work.

## 5 Evaluation

In this section, we present empirical evaluation results, which aim to answer the following questions:

**(RQ1)** Can CADET discover well-compressed structures for synthetic and real data?
**(RQ2)** How does the performance of CADET compare to prior work?
**(RQ3)** How useful are CDT-based enumeration and constraint-based ranking in the search?
**(RQ4)** Can the structures discovered by CADET generalizable to unseen data?

**General Experiment Setup** We run all the following experiments on a laptop with the Apple M3 Max CPU and 128 GB memory. In the experiments, we collect the running time and the compression ratio, which is defined as $\texttt{size}\left(\mathcal{X}\right)/\texttt{size}\left(\mathcal{R}_\mathcal{N}\right) = \texttt{size}\left(\mathcal{X}\right)/\sum_{v\in\mathcal{V}}\texttt{size}\left(v\right)$. We choose the top $k_\theta$ structure for actual decomposition from all enumerated candidates.

### 5.1 RQ1: Compression Performance of Cadet on Synthetic and Real Data

#### 5.1.1 Performance on Synthetic Data

**Experiment Setup** We evaluate CADET on synthetic data by generating random tensors contracted from random tree structures. This provides a basic test that CADET can identify structures with compression ratios equal to or better than the generation structure. Following prior work (Zheng et al., 2024), we generate order-4 tensors with dimensions $16 \times 18 \times 20 \times 22$ and order-5 tensors with dimensions $14 \times 16 \times 18 \times 20 \times 22$. Internal ranks are randomly sampled between 2 and 5. For each shape, we sample 50 structures with random tensor values, and contract them into data tensors.

Table 1: Performance of CADET on real datasets with error bounds $\varepsilon = 0.1$ and $\varepsilon = 0.01$. CR and RE represent the average compression ratio and reconstruction error; Mem is the peak memory usage in GB; the search time ($t_{tot}$, in seconds) is broken down into preprocessing ($t_{pre}$), constraint solving ($t_{cons}$), enumeration ($t_{enum}$), and final decomposition ($t_{decomp}$). Dataset dimensions: Light Field ($40 \times 60 \times 3 \times 9 \times 9$), BigEarthNet-Small ($30 \times 12 \times 120 \times 120$), BigEarthNet-Large ($5 \times 20 \times 30 \times 12 \times 120 \times 120$), and PDEBench ($10 \times 5 \times 21 \times 64 \times 64 \times 64$).

| Data | $\varepsilon$ | CR | RE | Search Time (s) | | | | | Mem (GB) |
|---|---|---|---|---|---|---|---|---|---|
| | | | | $t_{tot}$ | $t_{pre}$ | $t_{cons}$ | $t_{enum}$ | $t_{decomp}$ | |
| Light field | 0.1 | 68.81 | 0.099 | 17.49 | 0.00 | 17.29 | 0.15 | 0.04 | 0.43 |
| | 0.01 | 7.04 | 0.010 | 3.08 | 0.01 | 2.83 | 0.16 | 0.08 | |
| BigEarthNet-Small | 0.1 | 160.01 | 0.100 | 3.35 | 1.18 | 1.71 | 0.02 | 0.45 | 0.73 |
| | 0.01 | 2.67 | 0.010 | 8.82 | 3.18 | 1.38 | 0.06 | 4.20 | |
| BigEarthNet-Large | 0.1 | 151.79 | 0.100 | 4096.89 | 3898.37 | 124.16 | 5.33 | 69.00 | 26.75 |
| | 0.01 | 3.07 | 0.010 | 4640.90 | 3894.94 | 56.44 | 5.29 | 684.21 | |
| PDEBench | 0.1 | 38.75 | 0.100 | 2232.25 | 1817.40 | 341.03 | 3.71 | 70.09 | 16.77 |
| | 0.01 | 1.81 | 0.010 | 2654.70 | 1830.92 | 110.66 | 3.66 | 709.45 | |

**Result Analysis**  Fig. 7 compares the compression ratios of generation structures and those discovered by CADET. The results indicate that CADET achieves compression ratios that are equal to or greater than that of generation structures, for every data point. In other words, CADET identifies improved structures when the generation structure is suboptimal.

### 5.1.2 Performance on Real Data

**Experiment Setup**  To evaluate the performance of CADET on real-world data, we employ three datasets: light fields (Zheng et al., 2024), BigEarthNet (Sumbul et al., 2019), and PDEBench (Takamoto et al., 2022a;b). For the light field data, we use the bunny data with dimensions $40 \times 60 \times 3 \times 9 \times 9$, following prior work (Zheng et al., 2024). The BigEarthNet data is used to create tensors with dimensions $30 \times 12 \times 120 \times 120$ and $5 \times 20 \times 30 \times 12 \times 120 \times 120$ by randomly sampling and stacking 30 and 3000 data points respectively. For the PDEBench data, we sample 10 data points from the 3D compressible Navier-Stokes problem to create tensors of dimensions $10 \times 5 \times 21 \times 64 \times 64 \times 64$.

**Result Analysis**  We collect average compression ratios, search time in seconds, reconstruction errors, and peak memory usage for each dataset in Table 1. The results demonstrate that CADET achieves significant compression across diverse real-world datasets while strictly adhering to the prescribed error bounds. For an error tolerance of $\varepsilon = 0.1$, the algorithm yields high compression ratios, notably above 150 for the BigEarthNet datasets. As the error bound tightens to $\varepsilon = 0.01$, the compression naturally decreases, ranging from 1.81 to 7.04. The stable reconstruction errors across all benchmarks confirm that precomputed singular values provides an upper bound for real singular values: CADET uses precomputed singular values to search for rank assignments, and none of the numerical experiments witness the exceeding of the prescribed error bounds. Peak memory consumption increases with the tensor sizes but remains well within practical hardware limits. For the largest BigEarthNet-Large dataset, which has a disk footprint of approximately 2 GB, the algorithm's peak memory usage reaches 26.75 GB.

The overall search time ($t_{tot}$) scales with both the size and the order of data tensors. For low-dimensional, small data, the search time is usually below 20 second. For large data tensors, the search time rises up to around 5,000 second. An analysis of the time breakdown reveals that the computational bottleneck is heavily scale-dependent. For large-scale tensors, the runtime is overwhelmingly dominated by the precomputation of singular values ($t_{pre}$). This is mathematically expected, as the cost of SVDs increases with tensor size and the number of SVDs grows exponentially with tensor order. For small-scale tensors, constraint solving ($t_{cons}$)

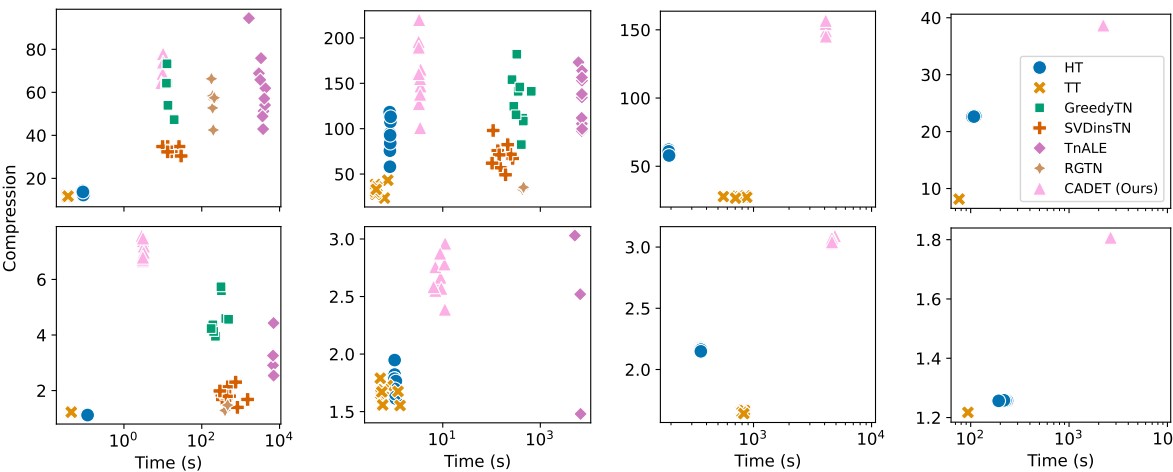

Figure 8: Comparison of compression ratio vs time on real datasets. The datasets from left to right are light field data ($40 \times 60 \times 3 \times 9 \times 9$), BigEarthNet-Small ($30 \times 12 \times 120 \times 120$), BigEarthNet-Large ($5 \times 20 \times 30 \times 12 \times 120 \times 120$), and PDEBench ($10 \times 5 \times 21 \times 64 \times 64 \times 64$). The two rows corresponds to error bounds of 0.1 and 0.01 respectively.

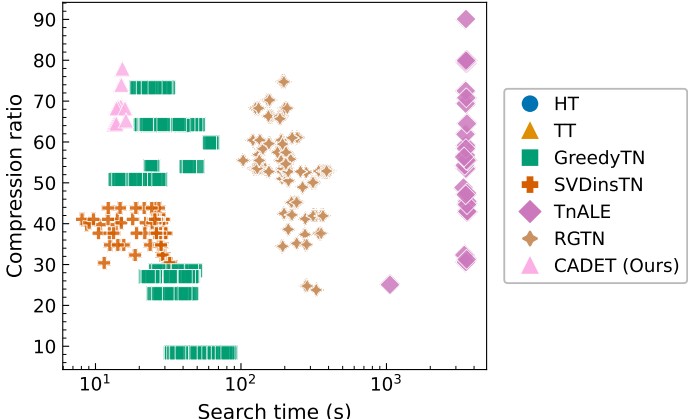

Figure 9: Compression Ratio vs. Runtime on light field data ($40 \times 60 \times 3 \times 9 \times 9$) with $\varepsilon = 0.1$ and a timeout limit 1 hour. Each data point of the baselines represents the structure search result for a data tensor with a specific hyperparameter configuration. The proposed method (CADET) achieves competitive compression while remaining orders of magnitude faster than high-performance baselines like TnALE.

represents the primary computational bottleneck. Notably, constraint solving requires longer execution time under looser error bounds, as a large error tolerance expands the feasible search space and increases the number of valid rank assignment configurations.

## 5.2   RQ2: Compression Performance Comparison Between Cadet and Existing Tools

**Experiment Setup**   We compare our method in terms of search time and compression ratio against tensor trains (TT), binary hierarchical tuckers (HT), and four baselines: (1) GreedyTN (Hashemizadeh et al., 2020): a greedy algorithm that gradually increases internal ranks; (2) TnALE (Li et al., 2023): a sampling-based method that reduces sample sizes by local search; (3) SVDinsTN (Zheng et al., 2024): an optimization-based approach that integrates rank and topology search. (4) RGTN (Wang et al., 2026): a multi-scale optimization approach that searches rank assignments and topology together. For our algorithm, we search dimension trees of up to 6 nodes and pick the top one for the tensor decomposition. The timeout limit is 3 hours.

**Result Analysis** The results are shown in Fig. 8. Compared to TT and HT, our algorithm finds more compressed structures, albeit requiring roughly an order of magnitude more time across all datasets. This is expected, as our approach explores general tree structures and allows reordering free indices, whereas TT and HT are limited to special cases within our broader search space.

In comparison to prior work, our method not only achieves comparable or superior compression ratios, especially on large tensors, but is also up to $10\times$ faster than baselines. While TnALE occasionally discovers structures with higher compression ratios by finding cyclic structures, its dependence on a large sample size leads to significant slowdowns across tests. Moreover, TnALE often fails to converge to the desired error bound within the timeout , and encounters out-of-memory issues with large tensors. TNGreedy, SVDinsTN, and RGTN exhibit performance similar to ours on smaller tensors and larger error bounds, but neither scales well as tensor size increases. They do not produce valid results within the 3-hour timeout for BigEarthNet-Small data with $\varepsilon = 0.01$ or other larger tensors with either error bound. To ensure these timeouts were not a consequence of suboptimal settings, we conducted extensive hyperparameter optimization for both SVDinsTN and RGTN, carefully tuning their configurations to strike the best possible balance between compression performance and computational efficiency.

To verify that the performance gains of CADET are consistent across different baseline parameterizations, we conducted an extensive parameter sweep on the light field dataset with $\varepsilon = 0.1$ and timeout limit of 1 hour. We tested a broad range of configurations for each baseline: rank boundaries from $[10, 15, 20]$ for TnALE; optimization steps from $[100, 200, 300]$, rank increments from $[1, 2, 4]$, and ALS iterations from $[100, 150, 200]$ for GreedyTN; the regularization parameters $\gamma \in [0.05, 0.1, 0.5]$, $\mu \in [0.1, 0.5, 1.0]$, and $\rho \in [10^{-4}, 10^{-5}, 10^{-6}]$ for SVDinsTN; the regularization parameters $\lambda_{re} \in [1.0, 2.0, 4.0]$, the expansion steps from $[15, 20, 25]$, the compression steps from $[15, 20, 25]$, and the initial dimension sizes from $[2, 4, 8]$. These parameter choices are centered around the manually tuned settings used in the previous experiment. Due to the significant runtime of the baselines, this exhaustive sweep was restricted to the light field data as a representative benchmark.

As shown in Fig. 9, the resulting performance trends mirror our previous evaluation in Fig. 8 (top left). CADET provides a superior balance of efficiency and compression performance, identifying high-quality structures with small search overhead. While TnALE can reach higher absolute compression ratios, largely due to its support for cyclic topologies, it requires orders of magnitude more search time and frequently hits the one-hour timeout. GreedyTN and RGTN exhibit significant performance variance across different parameter settings and data tensors, but no configuration reaches the compression performance of our method. SVDinsTN is more stable, but it frequently converges to suboptimal structures or fails to satisfy the error bound. In contrast, our approach CADET identifies high-quality tensor network structures without the need for the extensive hyperparameter tuning that limits the practical utility of existing methods.

These results confirm that our method outperforms baselines in terms of search time while maintaining matching or better compression ratios. Our method of deferring decomposition until the search space is narrowed down effectively reduces the search time. However, the lack of support for cycles limits our performance on compression ratio in some cases, and we plan to address this in future work.

## 5.3 RQ3: Ablation Studies

### 5.3.1 Dimension Tree Enumeration w/ vs w/o Canonicalization

We evaluate the effectiveness of canonical dimension tree enumeration by comparing its performance to general tree enumeration (removing the three checks on lines 9–11 of Algorithm 2). We set the maximum number of nodes in dimension trees to 6. We compare compression ratios and running time of the two settings on light field data with dimensions $40 \times 60 \times 3 \times 9 \times 9$, and BigEarthNet-Small data with dimensions $30 \times 12 \times 120 \times 120$. Ablation experiments on larger tensors did not produce results within 12 hours and are therefore omitted. Table 2 shows that both methods achieve similar compression ratios, but canonical dimension tree enumeration significantly accelerates the search process. For BigEarthNet-Small, allowing canonical checks results in only 63 CDTs whereas removing canonical checks leads to 35727 trees.

Table 2: Comparison of compression ratio (CR) and runtime on Light field data ($60 \times 40 \times 3 \times 9 \times 9$), and BigEarthNet-Small ($30 \times 12 \times 120 \times 120$) data across different variants of our algorithm. The variants differ based on the dimension tree enumeration method (canonical or non-canonical) and the rank search method (constraint-based or equal error distribution). Canonical + Constraint is ours. Results are shown for error bounds $\varepsilon = 0.1$ and $\varepsilon = 0.01$.

| Dataset | Variant | | $\varepsilon = 0.1$ | | $\varepsilon = 0.01$ | |
| --- | --- | --- | --- | --- | --- | --- |
| | Enum | Rank | CR | Time (s) | CR | Time (s) |
| Light field | Canonical | Constraint | 68.81 | **17.49** | 7.03 | **3.62** |
| | Non-canonical | Constraint | 68.34 | 4069.6 | 7.03 | 1635.2 |
| | Canonical | Equal | 69.18 | 28.73 | 7.03 | 44.32 |
| BigEarthNet-Small | Canonical | Constraint | 155.71 | **4.67** | 2.69 | **8.07** |
| | Non-Caonical | Constraint | 154.64 | 1248.06 | 2.66 | 419.65 |
| | Canonical | Equal | 147.51 | 15.04 | 2.62 | 68.97 |

Table 3: Comparison of tensor network sizes and compression ratios on the light field dataset ($\varepsilon = 0.1$) across three structures: the score from constraint solving, the final decomposition results achieved by CADET, and the best results from exhaustive rank enumeration. The estimated sizes closely approximate the final decomposition and approach the best size obtained from exhaustive enumeration. The Ranking in CADET column reports where the exhaustive-enumeration structure falls within CADET's own cost-sorted ranking, showing it is consistently ranked near the top. ($\downarrow$) and ($\uparrow$) means lower and higher is better respectively.

| Dataset | CADET | | | | Rank Enumeration | | |
| --- | --- | --- | --- | --- | --- | --- | --- |
| | Scoring | | Final | | Size ($\downarrow$) | CR ($\uparrow$) | Ranking by CADET ($\downarrow$) |
| | Size ($\downarrow$) | CR ($\uparrow$) | Size ($\downarrow$) | CR ($\uparrow$) | | | |
| Light Field | 10,007 | 58.28 | 9,038 | 64.53 | 7,240 | 80.55 | 11 |
| | 8,914 | 65.43 | 7,478 | 77.99 | 6,718 | 86.81 | 14 |
| | 10,007 | 58.28 | 9,038 | 64.53 | 7,448 | 78.30 | 13 |
| | 9,038 | 64.53 | 8,478 | 68.79 | 6,747 | 86.44 | 13 |
| | 9,727 | 59.96 | 8,518 | 68.47 | 7,182 | 81.20 | 9 |
| | 9,065 | 64.34 | 7,878 | 74.03 | 6,798 | 85.79 | 13 |
| | 10,007 | 58.28 | 9,038 | 64.53 | 7,448 | 78.30 | 6 |
| | 9,514 | 61.30 | 8,545 | 68.25 | 7,240 | 80.55 | 12 |
| | 9,514 | 61.30 | 8,545 | 68.25 | 6,990 | 83.43 | 9 |
| | 10,094 | 57.78 | 8,934 | 65.28 | 7,456 | 78.22 | 8 |
| Mean | 9,589 | 60.95 | 8,549 | 68.46 | 7,127 | 81.96 | 10.8 |

### 5.3.2 w/ vs w/o Constraint-Based Rank Search

To evaluate the effectiveness of constraint-based rank search, we compare it against a variants that enumerates all possible rank assignments for a tree. Figure 10 presents the changes of compression ratios and search time for the 10 light field data tensors at $\varepsilon = 0.1$. It shows that our proposed method reduces the search time by approximately two orders of magnitude while maintaining high compression ratios, typically reaching 80–90% of the values found by the exhaustive rank-enumeration method.

The quality of constraint solving results is further detailed in Table 3, where we compare the estimated network size through constraint solving, the network size obtained in final decomposition, and the network size achieved from enumerative rank search. Across the 10 light field data tensors, the structural size estimated via constraint solving closely mirrors the final decomposition. On average, the constraint solver yields an estimated tensor network size of 9,589 (CR = 60.95), which the rounding procedure in CADET refines to a final decomposition size of 8,549 (CR = 68.46). This final output closely approaches the optimum

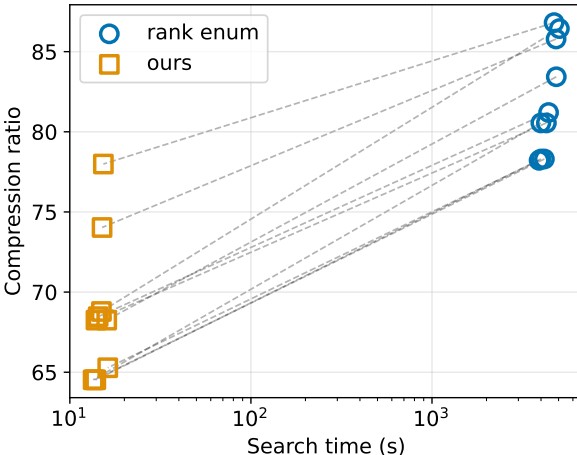

Figure 10: Performance comparison between constraint-based and rank-enumeration-based scoring for light field data. The proposed constraint-based method reduces search time by approximately two orders of magnitude at the cost of a moderate reduction in compression ratio (CR), which remains within 80-85% of the rank-enumeration method. Dotted lines indicate corresponding data points between two methods.

Table 4: Performance of network search on the training batch, the discovered best tree structure on the test batches, and hierarchical Tuckers (HT) on the test batch with respect to compression ratios and time.

|  | BigEarthNet-Small | | PDEBench | |
|---|---|---|---|---|
|  | CR | Time (s) | CR | Time (s) |
| Train | 157.12 | 4591.64 | 38.75 | 2141.85 |
| Test | 148.98 | 109.61 | 38.58 | 115.38 |
| HT | 60.71 | 190.98 | 22.65 | 95.21 |

obtained via exhaustive rank enumeration, which yields a mean size of 7,127 (CR = 81.96). To quantify the gap between our constraint-based scoring and exhaustive enumeration, we evaluate where the structure selected by exhaustive rank enumeration falls within CADET's own cost-sorted candidate ranking. Notably, this optimal tree structure consistently ranks among the top candidates, with a mean rank of 10.8 (ranging from 6 to 14). These findings demonstrate that our constraint-based rank search effectively bypasses the combinatorial explosion of exhaustive search while still identifying highly competitive rank assignments.

However, it is unaffordable to run exhaustive rank enumeration on light field data with $\varepsilon = 0.01$ or BigEarthNet data with either error tolerance, we develop a cheaper variant that distributes the error budget equally between steps, which is the strategy adopted by HT and TT. As shown in Table 2, this equal-error variant exhibits longer execution times than our constraint-solving approach. Although it occasionally yields slightly superior compression ratios due to its execution of online tensor decomposition, CADET relies on approximated singular values, which can lead to more conservative rank estimations but significantly faster runtimes. Such reduction of computation time is especially evident for small error bounds.

### 5.4 RQ4: Generalization Analysis

#### 5.4.1 Generalization to Unseen Data

**Experiment Setup** We conduct a generalization test to evaluate if the structure discovered by our algorithm generalizes to unseen data. To begin, each dataset is divided into equal-sized batches. The first batch serves as the training batch, where both topology and rank search are performed using the proposed algorithm. For subsequent test batches, we apply the topology from the training batch, and run the constraint-

based rank search to assess generalization. In the BigEarthNet dataset, data is divided into 89 batches of tensors with shape $5 \times 20 \times 30 \times 12 \times 120 \times 120$, each containing 3000 samples. In the PDEBench dataset, the data is divided into 60 batches of 10 samples each, producing tensors of shape $10 \times 5 \times 21 \times 64 \times 64 \times 64$. The error bound used for this experiment is $\varepsilon = 0.1$.

**Result Analysis** Table 4 presents the compression ratios and running time for the training batch and the averages of test batches. The results show that the compression ratios of test batches are close to that of the training batch, which proves that the discovered topology generalizes well to unseen data. Particularly, although the training batch requires thousands of seconds to find the optimal tree structure within the search space, it takes only about 110 seconds on average to perform the decomposition for each test batch. This result reinforces the feasibility of running the training batch once to determine the best tree topology and reusing it for subsequent batches, which significantly reduces the overall computation time.

## 6 Conclusion

In this work, we introduced CADET, a framework for tensor network structure search based on canonical dimension trees. By representing potential architectures as hierarchical index partitions, we successfully decouple topology enumeration from rank optimization, which significantly reduces the complexity of the search space. Our methodology replaces the traditional, computationally expensive candidate assessment methods with an efficient constraint-solving mechanism. By precomputing a metadata map of singular values, we solve for rank assignments that are empirically close to the configurations found by exhaustive enumeration, and evaluate candidate quality without performing live tensor decompositions for every structure.

Our empirical results on both synthetic and real-world datasets demonstrate that the proposed approach consistently discovers well-compressed structures, achieves competitive compression ratios compared to existing tools, and scales to larger tensors where prior methods fail. Moreover, the structures identified by our method generalize to unseen data from the same source, enabling efficient reuse in practical applications.

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

## A  Proofs

### A.1  Proof of Singular Value Approximations

**Definition A.1** (Subtensor). Suppose $\mathcal{X} \in \mathbb{R}^{n_1 \times n_2 \times \dots n_d}$ is a $d$-dimensional tensor. Its subtensor $\mathcal{Y} \in \mathbb{R}^{m_1 \times m_2 \times \dots m_d}$ (written as $\mathcal{Y} \sqsubseteq \mathcal{X}$) is also a $d$-dimensional tensor obtained by restricting the index sets corresponding to dimension $\mu$ to $m_\mu$ elements where for all $\mu \in \{1, 2, \dots, d\}$, $m_\mu \leq n_\mu$. If $\mathcal{Y}$ is a subtensor of $\mathcal{X}$, there exists a set of binary matrices with orthonormal rows $\pi_1, \dots \pi_d$ such that $\mathcal{Y} = (\pi_1 \otimes \pi_2 \otimes \dots \otimes \pi_d)\mathcal{X}$.

**Lemma A.2** (Subtensors Preservation on Permutation). *Let $\mathcal{X} \in \mathbb{R}^{n_1 \times n_2 \times \dots \times n_d}$ be a $d$-dimensional tensor, $\mathcal{Y} \in \mathbb{R}^{m_1 \times m_2 \times \dots \times m_d}$, and $\mathcal{Y} \sqsubseteq \mathcal{X}$. If $\Pi \in \{1, \dots, d\} \to \{1, \dots, d\}$ is a permutation of the dimensions, then* $\texttt{permute}(\mathcal{Y}, \Pi(1), \dots, \Pi(d)) \sqsubseteq \texttt{permute}(\mathcal{X}, \Pi(1), \dots, \Pi(d))$.

*Proof.* As $\mathcal{Y} \sqsubseteq \mathcal{X}$, there exists a list of binary matrices with orthonormal rows $[\pi_\mu]_{1 \leq \mu \leq d}$ such that $\mathcal{Y} = (\pi_1 \otimes \pi_2 \otimes \dots \otimes \pi_d)\mathcal{X}$. Then, we can see that

$$
\begin{aligned}
&\texttt{permute}\,(\mathcal{Y}, \Pi(1), \Pi(2), \dots, \Pi(d)) \\
=\quad &\texttt{permute}\,((\pi_1 \otimes \pi_2 \otimes \dots \otimes \pi_d)\mathcal{X}, \Pi(1), \Pi(2), \dots, \Pi(d)))
\end{aligned}
\tag{7}
$$

Since each $\pi_\mu$ only modifies values along the corresponding mode $\mu$, and permutation only moves dimensions without altering values, we can move the projection outside the $\texttt{permute}$ by reordering the projections according to the dimension permutation, and get

$$
\begin{aligned}
&\texttt{permute}\,((\pi_1 \otimes \pi_2 \otimes \dots \otimes \pi_d)\mathcal{X}, \Pi(1), \Pi(2), \dots, \Pi(d))) \\
=\quad &(\pi_{\Pi(1)} \otimes \pi_{\Pi(2)} \otimes \dots \otimes \pi_{\Pi(d)})\,\texttt{permute}\,(\mathcal{X}, \Pi(1), \Pi(2), \dots, \Pi(d))
\end{aligned}
\tag{8}
$$

Therefore, $\texttt{permute}(\mathcal{Y}, \Pi(1), \Pi(2), \dots, \Pi(d)) \sqsubseteq \texttt{permute}(\mathcal{X}, \Pi(1), \Pi(2), \dots, \Pi(d))$ holds. $\qquad\square$

**Lemma A.3** (Subtensors Preservation on Reshape). *Let $\mathcal{X} \in \mathbb{R}^{n_1 \times n_2 \times \dots n_d}$ be a $d$-dimensional tensor with indices $I_1, I_2, \dots, I_d$, and $\mathcal{Y} \in \mathbb{R}^{m_1 \times m_2 \times \dots m_d}$ be a $d$-dimensional tensor with indices $J_1, J_2, \dots, J_d$. If $\mathcal{Y} \sqsubseteq \mathcal{X}$, then for any $s \subset \{1, 2, \dots, d\}$, we have $\mathcal{X}^{(\mathcal{I}_s)} \sqsubseteq \mathcal{Y}^{(\mathcal{J}_s)}$, and vice versa.*

*Proof.* Since $\mathcal{Y} \sqsubseteq \mathcal{X}$, there exists binary matrices with orthonormal rows $[\pi_\mu]_{\mu \in \{1, 2, \dots, d\}}$ such that

$$
\mathcal{Y} = (\pi_1 \otimes \pi_2 \otimes \dots \otimes \pi_d)\,\mathcal{X}
\tag{9}
$$

For a set of indices $\mathcal{I}_s$, by Theorem A.2 and the associativity of tensor product, we get that

$$
\mathcal{Y}^{(\mathcal{J}_s)} = \left( \bigotimes_{\mu \in s} \pi_\mu \otimes \bigotimes_{\mu \notin s} \pi_\mu \right) \mathcal{X}^{(\mathcal{I}_s)}
\tag{10}
$$

Therefore, we have that $\mathcal{Y}^{(\mathcal{J}_s)} \sqsubseteq \mathcal{X}^{(\mathcal{I}_s)}$.

Similarly, the property can be proved for the reverse direction by applying the above two steps in the reverse order. $\qquad\square$

**Lemma A.4** (Singular Value Upper Bound in Subtensors). *Let $\mathcal{X} \in \mathbb{R}^{n_1 \times n_2 \times \dots \times n_d}$ be a $d$-dimensional tensor with indices $I_1, I_2, \dots, I_d$, $\mathcal{Y} \in \mathbb{R}^{m_1 \times m_2 \times \dots \times m_d}$ be a $d$-dimensional tensor with indices $J_1, J_2, \dots, J_d$, and $\mathcal{Y} \sqsubseteq \mathcal{X}$. Define $\sigma_i(A)$ to be the $i^{th}$ largest singular value of a matrix $A$. Then, for all $s \subset \{1, 2, \dots, d\}$, if $\mathcal{I}_s = \{I_i\}_{i \in s}$ and $\mathcal{J}_s = \{J_i\}_{i \in s}$, we have $\sigma_i(\mathcal{Y}^{(\mathcal{J}_s)}) \leq \sigma_i(\mathcal{X}^{(\mathcal{I}_s)})$.*

*Proof.* From $\mathcal{Y} \sqsubseteq \mathcal{X}$ and Theorem A.2, we know that $\mathcal{Y}^{(\mathcal{J}_s)}) \sqsubseteq \mathcal{X}^{(\mathcal{I}_s)}$. Then, the result can be obtained by applying the Poincaré separation theorem to $\mathcal{X}^{(\mathcal{I}_s)}(\mathcal{X}^{(\mathcal{I}_s)})^*$. $\qquad\square$

**Lemma A.5** (Singular Value Upper Bound in Truncations). *Let $\mathcal{X} \in \mathbb{R}^{n_1 \times n_2 \times \dots \times n_d}$ be a $d$-dimensional tensor with indices $I_1, I_2, \dots, I_d$. Let $\mathcal{I}_s \subseteq \{I_1, I_2, \dots, I_d\}$ be a set of indices. Suppose $SVD(\mathcal{X}^{(\mathcal{I}_s)}) = U\Sigma V$. After truncation to some rank $r$, we get $\widetilde{\mathcal{X}}^{(\mathcal{I}_s)} = \widetilde{U}\widetilde{\Sigma}\widetilde{V}$ where $\widetilde{U} = U[:, :r]$, $\widetilde{\Sigma} = \Sigma[:r, :r]$, and $\widetilde{V} = V[:r]$. Then we have that, for all $\mathcal{I}_t \subset \{I_1, I_2, \dots, I_d\}$, if $\mathcal{I}_t \subseteq \mathcal{I}_s, \mathcal{I}_s \subseteq \mathcal{I}_t$, or $\mathcal{I}_s \cap \mathcal{I}_t = \emptyset$, then $\sigma_i(\widetilde{\mathcal{T}}^{(\mathcal{I}_t)}) \leq \sigma_i(\mathcal{X}^{(\mathcal{I}_t)})$.*

*Proof.* In this proof, matrices $U$ and $V$ can be treated as $(n_s + 1)$ and $(d - n_s + 1)$ dimensional tensors where $n_s = |\mathcal{I}_s|$. The same considerations are made for $\widetilde{U}$ and $\widetilde{V}$. Due to the tree structure, we consider the following three cases of relations between $\mathcal{I}_t$ and $\mathcal{I}_s$.

- Case I: $\mathcal{I}_t = \mathcal{I}_s$. The singular values are discarded without other modification, so $\sigma_i(\widetilde{\mathcal{X}}^{(\mathcal{I}_t)}) = \sigma_i(\mathcal{X}^{(\mathcal{I}_t)})$.

- Case II: $\mathcal{I}_t \subset \mathcal{I}_s$. By the definition of matricization and SVD, we know that $\mathcal{I}_s \subset \text{INDICES}(\widetilde{U})$. Hence, $\mathcal{I}_t \subset \text{INDICES}(\widetilde{U})$. By Theorem A.3, Theorem A.4, and $\widetilde{U}\widetilde{\Sigma} \sqsubseteq U\Sigma$, we know that

$$\sigma_i\left(\left(\widetilde{U}\widetilde{\Sigma}\right)^{(\mathcal{I}_t)}\right) \leq \sigma_i\left((U\Sigma)^{(\mathcal{I}_t)}\right) \tag{11}$$

Therefore,

$$\sigma_i\left(\widetilde{\mathcal{X}}^{(\mathcal{I}_t)}\right) = \sigma_i\left(\left(\widetilde{U}\widetilde{\Sigma}\right)^{(\mathcal{I}_t)}\right) \leq \sigma_i\left((U\Sigma)^{(\mathcal{I}_t)}\right) = \sigma_i\left(\mathcal{X}^{(\mathcal{I}_t)}\right) \tag{12}$$

- Case III: $\mathcal{I}_s \subset \mathcal{I}_t$ or $\mathcal{I}_t \cap \mathcal{I}_s = \emptyset$. By the definition of matricization and SVD, we know that $\mathcal{I}_s \subset \text{INDICES}(\widetilde{U})$. Hence, $\mathcal{I}_t \subset \text{INDICES}(\widetilde{\mathcal{X}}) \setminus \text{INDICES}(\widetilde{U}) \subset \text{INDICES}(\widetilde{V})$. By Theorem A.3, Theorem A.4, and $\widetilde{\Sigma}\widetilde{V} \sqsubseteq \Sigma V$, we know that

$$\sigma_i\left((\widetilde{\Sigma}\widetilde{V})^{(\mathcal{I}_t)}\right) \leq \sigma_i\left((\Sigma V)^{(\mathcal{I}_t)}\right) \tag{13}$$

Therefore,

$$\sigma_i\left(\widetilde{\mathcal{X}}^{(\mathcal{I}_t)}\right) = \sigma_i\left(\left(\widetilde{\Sigma}\widetilde{V}\right)^{(\mathcal{I}_t)}\right) \leq \sigma_i\left((\Sigma V)^{(\mathcal{I}_t)}\right) = \sigma_i\left(\mathcal{X}^{(\mathcal{I}_t)}\right) \tag{14}$$

To summarize, we have $\sigma_i\left(\widetilde{\mathcal{X}}^{(\mathcal{I}_t)}\right) \leq \sigma_i\left(\mathcal{X}^{(\mathcal{I}_t)}\right)$ for all possible choices of $\mathcal{I}_t$. $\square$

**Theorem A.6** (Singular Value Upper Bound). *Let $\mathcal{X} \in \mathbb{R}^{n_1 \times \cdots \times n_d}$ be a d-dimensional tensor, compressing $\mathcal{X}$ into a structure described by a dimension tree $T_{\mathcal{I}}$ of $n + 1$ nodes $\mathcal{I}, \mathcal{I}_1, \mathcal{I}_2, \ldots, \mathcal{I}_n$ produces the structure $\mathcal{N}$, then for every $1 \leq i, s \leq n$, we have $\sigma_j\left(\mathcal{R}_{\mathcal{N}_i}^{(\mathcal{I}_s)}\right) \leq \sigma_j\left(\mathcal{X}^{(\mathcal{I}_s)}\right)$ where $\sigma_j(A)$ is the $j^{th}$ largest singular value of the matrix $A$, and $\mathcal{N}_i$ is the network obtained after the first $i$ tensor decompositions specified by $T_{\mathcal{I}}$.*

*Proof.* By the definition of dimension trees, for every pair $1 \leq s < t \leq n$, there could only be three relations between $\mathcal{I}_s$ and $\mathcal{I}_t$: $\mathcal{I}_s \subset \mathcal{I}_t$, $\mathcal{I}_t \subset \mathcal{I}_s$, or $\mathcal{I}_s \cap \mathcal{I}_t = \emptyset$.

Suppose the network obtained after the $k^{th}$ tensor decomposition is denoted as $\mathcal{N}_k$. The network obtained after performing the tensor decomposition on $\mathcal{N}_k$ along index set $\mathcal{I}_k$ is $\mathcal{N}_{k+1}$. Performing the split defined above is equivalent to performing a truncated SVD on $\mathcal{R}_{\mathcal{N}_k}^{(\mathcal{I}_k)}$. Formally, we can say that if $\mathcal{R}_{\mathcal{N}_k}^{(\mathcal{I}_k)} = U\Sigma V$, then $\mathcal{R}_{\mathcal{N}_{k+1}}^{(\mathcal{I}_k)} = \widetilde{U}\widetilde{\Sigma}\widetilde{V}$, where $\widetilde{U}$, $\widetilde{\Sigma}$, and $\widetilde{V}$ are truncated matrices of $U$, $\Sigma$, and $V$. Consequently, using Theorem A.5, we have that, for $\mathcal{I}_t \subset \{I_1, \ldots, I_d\}$ such that $\mathcal{I}_t \subseteq \mathcal{I}_s, \mathcal{I}_s \subseteq \mathcal{I}_t$, or $\mathcal{I}_s \cap \mathcal{I}_t = \emptyset$, $\sigma_i\left(\mathcal{R}_{\mathcal{N}_{k+1}}\right) \leq \sigma_i\left(\mathcal{R}_{\mathcal{N}_k}\right)$ for all possible $i$ and $k$.

From the above result, we can conclude that $\sigma_i\left(\mathcal{R}_{\mathcal{N}_k}^{(\mathcal{I}_t)}\right) \leq \sigma_i\left(\mathcal{R}_{\mathcal{N}_0}^{(\mathcal{I}_t)}\right) = \sigma_i\left(\mathcal{X}^{(\mathcal{I}_t)}\right)$ for all valid choices of $\mathcal{I}_t$ and all possible values of $i$ and $k$. $\square$

**Theorem A.7** (Upper Bound of Costs). *For a data tensor $\mathcal{X}$ with indices $\mathcal{I}$, a dimension tree $T_{\mathcal{I}}$ of $n + 1$ nodes $\mathcal{I}_0 = \mathcal{I}, \mathcal{I}_1, \mathcal{I}_2, \ldots, \mathcal{I}_n \subset \mathcal{I}$, and an error bound $\varepsilon$, let $\sigma_{si}$ be the $i^{th}$ largest singular value of $\mathcal{X}^{(\mathcal{I}_s)}$, and $\varsigma_{si}$ be the $i^{th}$ largest singular value of $\hat{\mathcal{X}}^{(\mathcal{I}_s)}$ where $\hat{\mathcal{X}}$ is obtained after performing truncated SVD over $\mathcal{X}$. If $r_1, \ldots, r_n$ is a solution to the constraint solving with singular values $\sigma_{si}$ for all $\mathcal{I}_s \subset \mathcal{I}$, then $r_1, \ldots, r_n$ is also a solution to the constraint solving with singular values $\varsigma_{si} \leq \sigma_{si}$.*

*Proof.* It is easy to see that

$$\sum_{s=1}^{n} \sum_{i > r_s} \varsigma_{si}^2 \leq \sum_{s=1}^{n} \sum_{i > r_s} \sigma_{si}^2 \leq \left(\varepsilon \|\mathcal{X}\|_F\right)^2$$

which also satisfies the linear programming constraints. $\square$

## B  Example Structures

This section presents structures discovered by our tool and features the necessity of TN-SS. In each structure, square nodes denote free indices, labeled in the format of "<index name>-<index size>". Round nodes represent the constituent tensors, which are linked by contraction edges annotated with respective ranks.

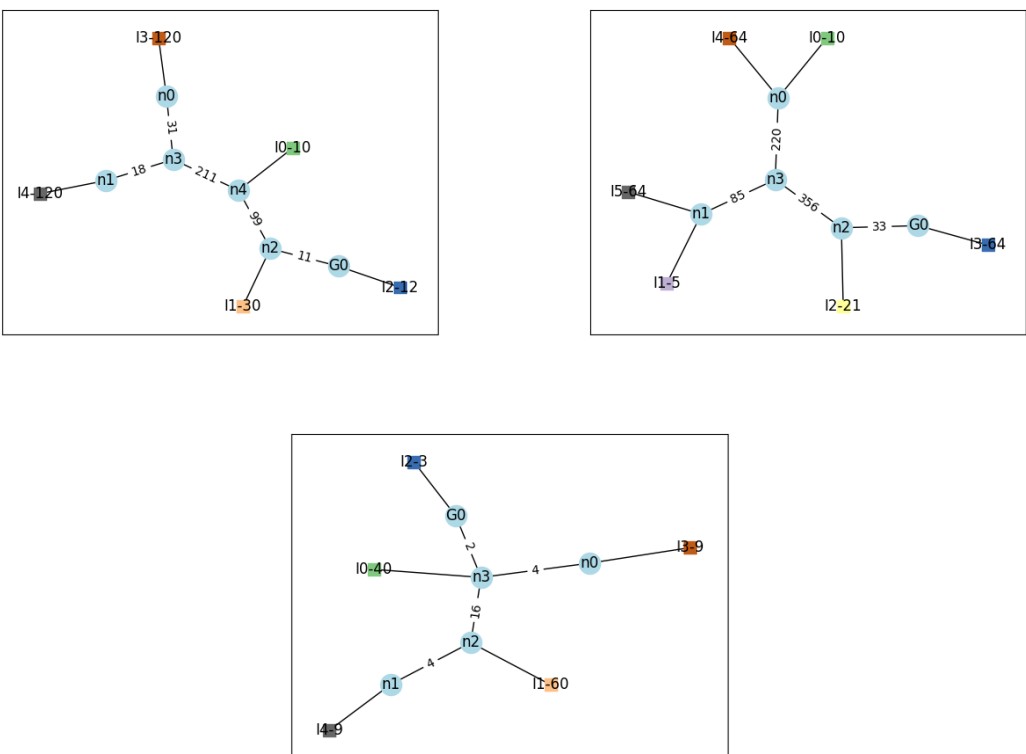

Figure 11: These structures showcase that our tool can discover non-standard structures other than TT, HT, etc. One single internal node suffices to provide good compression ratio in these cases.

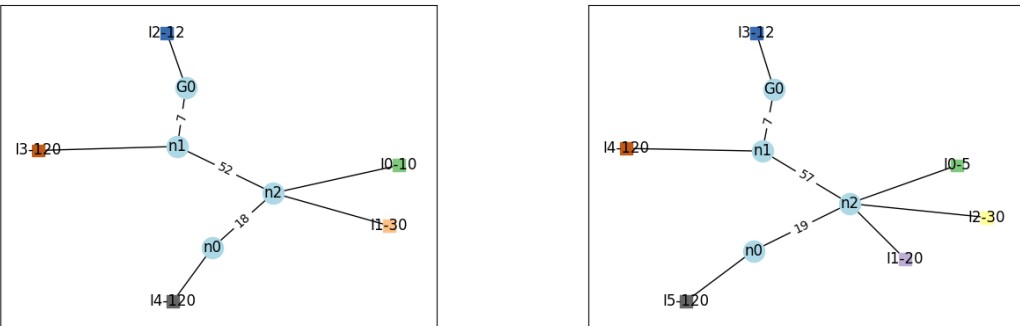

Figure 12: These two structures are similar to tensor trains but they have clustered and reordered indices, which allow them to have better compression ratios than traditional tensor trains.

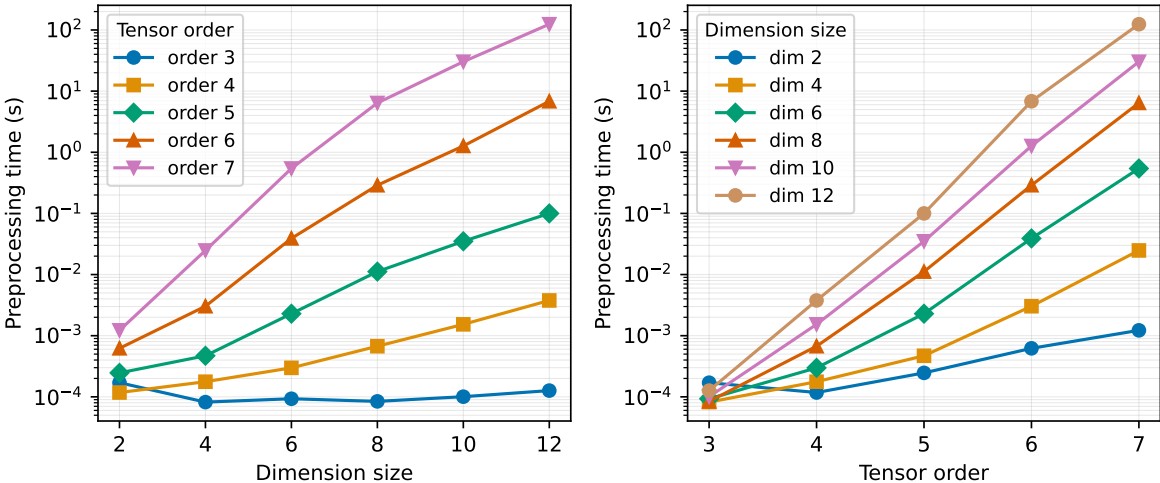

Figure 13: Preprocessing time (in seconds) on synthetic tensors using standard double-precision floating-point arithmetic. Left: Preprocessing time as a function of mode (dimension) size across different tensor orders ($d = 3$ to $7$). Right: Preprocessing time as a function of tensor order across different mode sizes ($n = 2$ to $12$). Both plots use a logarithmic scale for the vertical axis to highlight the scaling characteristics.

## C    Additional Experiment Results

### C.1    Preprocessing Complexity and Empirical Boundaries

To complement the real-world dataset benchmarks presented in Table 1, we isolate the practical scalability boundaries of CADET's preprocessing phase using controlled synthetic tensors. The preprocessing phase primarily involves constructing the $\Omega$ matrix by computing the singular values for all $2^{d-1}$ feasible matricizations.As illustrated in Figure 13 (Left), when the tensor order ($d$) is fixed, the wall-clock time scales predictably with the underlying matrix dimensions passed to the standard SVD solvers. Conversely, Figure 13 (Right) underscores the exponential impact of the tensor order $d$ on the preprocessing footprint. For moderate-order tensors ($d \leq 5$) with typical mode sizes, the preprocessing time remains negligible or completes within a few seconds. For higher-order configurations (e.g., $d = 7, n = 12$), the exponential growth term ($\mathcal{O}(n^{1.5d}2^d)$) becomes prominent, pushing wall-clock times past $10^2$ seconds.These empirical trends justify our focus on moderate-order tensors (up to order 6) in the main evaluation. Because this computational overhead is a one-time setup cost that replaces the expensive, iterative global decompositions required by cyclic topology search baselines, it remains a highly practical and advantageous trade-off for the target empirical regime.

