# OpenReview forum: "Tensor Network Structure Search Via Canonical Dimension Tree Enumeration"
_TMLR — Under review for TMLR_

### Review · Reviewer_cz14 · 2026-05-29

**Summary Of Contributions:**

The paper presents a clear and practically useful approach to speeding up tensor network structure search for compression, through an SVD-based proxy that allows candidate structures to be scored without repeatedly performing full (expensive) tensor decompositions each time. In my understanding, its primary contributions lie in the proposed canonical dimension-tree representation for reducing redundant enumeration of different possible topologies, and a rank-assignment procedure that uses precomputed singular values to efficiently identify promising tensor network structures.

# Strengths

One key strength of the paper is its clear technical formulation: the authors do a good job of concisely introducing the technical foundations of tensor networks and canonical dimension trees in enough detail to make the main content digestible, but without overloading the paper.

More importantly, I also find the authors' motivation compelling; removing the need to pay the cost of fully evaluating each candidate solution in the search space. This seems like an interesting and practically useful contribution, and the SVD-based approximation appears sound and well-motivated.

# Weaknesses

The primary weaknesses in my view lie in the mixed empirical support of the paper's claims, and positioning with respect to related work. The former is discussed in the next section. For the latter, the discussion of the related work in "Tensor Network Structure Search" seems incomplete. The most recent paper discussed is from 2024 here. While I understand that TMLR is not primarily novelty-driven, it's nonetheless important to discuss more recent developments over the last 2 years to better situate the proposed method in relation to the literature.

Specifically, the authors cite [1] in Sect 3.3.1. From what I can tell, the works seem to address a very similar problem on tensor network structure search, with a very similar motivation. I think it should be discussed more substantively in the related-work section. The authors should clarify the technical relationship between Cadet and this hierarchical version, including which limitations of the present method it addresses, whether the methods are directly comparable, and whether an experimental comparison is appropriate. If a direct comparison is not meaningful, the paper should explicitly explain why explicitly.

Furthermore, there are at least two more recent papers that the authors might consider discussing in Section 1.2 [2,3], and possibly comparing to if the experimental settings are compatible. I would also suggest a more thorough discussion of existing surveys on tensor methods in Section 1.2 to help non-specialist readers, for example [4,5].

---

* [1] Guo, Zheng, et al. "Hierarchical Tensor Network Structure Search for High-Dimensional Data." arXiv preprint arXiv:2603.27856 (2026).
* [2] Wang, Maolin, et al. "Renormalization Group Guided Tensor Network Structure Search." Proceedings of the AAAI Conference on Artificial Intelligence. Vol. 40. No. 31. 2026.
* [3] Iacovides, Giorgos, et al. "Domain-aware tensor network structure search." arXiv preprint arXiv:2505.23537 (2025).
* [4] Kolda, Tamara G., and Brett W. Bader. "Tensor decompositions and applications." SIAM review 51.3 (2009): 455-500.
* [5] Panagakis, Yannis, et al. "Tensor methods in computer vision and deep learning." Proceedings of the IEEE 109.5 (2021): 863-890.

**Audience:**

Yes

**Audience Explanation:**

I believe so. I am not an expert in tensor network-based compression, but I believe the SVD-based approximation would be of interest to many readers given its practical gains. This is especially true if authors can further show, practically, to what extent and if/when it can recover close to the optimal solutions in the oracle-style ablation.

**Broader Impact Concerns:**

None of note.

**Claims And Evidence:**

No

**Claims Explanation:**

The paper provides convincing evidence that the proposed Cadet can substantially reduce search time and can find practically useful compressed tensor network structures. However, I do not think all of the strongest claims are fully supported as currently written. In particular, I see two issues with specific key claims, and supporting experimental evidence:

# compression-ratio claims

The authors claims in the abstract are fairly strong: "Experimental results show that our approach improves search speed by up to 10× and achieves compression ratios 1.5× to 3× better than state-of-the-art". This claim to me reads as if improvements are made both to speed, and to compression ratios simultaneously. Furthermore, towards the end of Section 5.2, the authors write that "These results confirm that our method outperforms baselines in terms of search time and compression ratio.".

However, I do not think that the results in Figures 8 & 9 support these claims, as written currently. It is certainly true the experimental results show that the proposed algorithm is faster than TnALE. However, as is clear in Fig 8.1, 8.6, and Fig 9, the TnALE baseline achieves up to twice the compression rate of the proposed cadet. Therefore, I do not think the results are competitive on compression, despite being cheaper. I think the authors should revise the claims accordingly to emphasize the speed-performance trade-off. This appears to be the key conclusion of the results in my opinion. For example, there are clear strengths here over GreedyTN, along the speed axis that could be emphasized instead.  This does not undermine the practical value of the method (especially if the goal is efficient search under realistic computational budgets). However, I think the authors should revise the wording of the claims to more accurately reflect the results. For example, the paper could emphasize that Cadet achieves strong or competitive compression ratios at substantially lower search cost, and that it scales better than several baselines. If the authors want to claim superior compression ratios, this should be qualified by dataset, baselines, topology family (e.g. cyclic or not).

# singular value pre-computation could be validated against an oracle evaluation

One of the key contributions of the authors' work is in the precomputation of the singular values of various matrix unfoldings, meaning one does not need to compute the full decomposition for every candidate tensor network configuration. This is a useful idea, and well-motivated.

However, the authors also make claims about finding near-optimal ranks in this context ("This allows us to solve for the near-optimal ranks"), but speed alone does not validate the approximation as a good search/scoring proxy.

I think the paper would benefit from an “oracle” ablation on a smaller tensor where exhaustive evaluation is feasible. For example, the authors could compare their proposed scoring method against an oracle that runs the full decomposition for every candidate topology (or at least for all candidates in a restricted search space). This would allow them to report how often the SVD proxy-chosen candidate actually matches the true best candidate, and the compression ratio difference between the solution found via the proxy and the oracle solution.

This would strengthen the empirical support for the proposed approximation, and add weight to their claim about finding near-optimal ranks. At present, the method is clearly faster, but it is less clear how close the proxy-guided search is to the solution that would be obtained by full candidate evaluation.

**Requested Changes:**

Major:
* **Revise the compression-ratio claims**: The current wording suggests Cadet improves both search time and compression ratio over state-of-the-art baselines. The results appear more nuanced; cadet is clearly faster and often competitive, but does not uniformly achieve the best compression ratios. The claims should be qualified accordingly.
* **Add an oracle-style validation of the proposed SVD-based scoring proxy**: Since SVD-based precomputation is central to the method, the authors should test how well the proxy-guided search matches full candidate evaluation on a small tensor or restricted search space.
* **Update the related work**: The Tensor Network Structure Search discussion appears to stop at 2024. The authors should discuss relevant 2025-2026 work, or explain why it is not directly comparable.

Minor, but still important:

* **Citation style**: Citations should use \citep{} in the introduction and throughout the paper; currently they are written as inline text (presumably with \citet{} instead).
* **Supporting claims in introduction**: Some claims are made without supporting evidence: such as "A significant limitation of existing TN-SS methods is the redundant enumeration of network topologies alongside rank configurations." in Sect. 1.1 -- a critical discussion of why specific related work comes with this limitations is useful for situating the method in the literature
* **Citing prior work around definitions**: When introducing tensor networks around the Defn. 2.4., the authors should make it clear through citations what prior work, definitions, or formulations they are appealing to.
* **Bloat in introduction**: The subsection "1.1. Efficient TN-SS via Canonical Dimension Tree Enumeration" of the introduction contains far too many low-level details of the proposed method before introducing the necessary pre-requisites. I would suggest that readers will find the work more digestible if this introduction described the contributions more briefly in the first instance.

---

> ### Author Response · Authors · 2026-07-04
> **Response to Reviewer cz14**
>
> > Q1: Revise the compression-ratio claims: The current wording suggests Cadet improves both search time and compression ratio over state-of-the-art baselines. The results appear more nuanced; cadet is clearly faster and often competitive, but does not uniformly achieve the best compression ratios. The claims should be qualified accordingly.
>
> We completely agree with the reviewer’s assessment and have qualified our compression-ratio claims throughout the revised manuscript (including the Abstract and Evaluation sections). We now explicitly state that CADET achieves comparable and competitive compression ratios while delivering significantly shorter search times, rather than uniformly outperforming all baselines in compression.
>
> Additionally, during a thorough audit of our evaluation pipeline, we identified and corrected an implementation detail in the baseline TnALE script: the baseline's compression ratio calculation previously omitted index permutations, relying solely on the adjacency matrix. After correcting this to fully account for index permutations, TnALE occasionally achieves up to ~15% better compression than CADET due to its utilization of cyclic structures. However, CADET's compression performance remains highly competitive and within a similar range, while offering orders-of-magnitude faster search times.
>
> > Q2: Add an oracle-style validation of the proposed SVD-based scoring proxy: Since SVD-based precomputation is central to the method, the authors should test how well the proxy-guided search matches full candidate evaluation on a small tensor or restricted search space.
>
> We included exactly this type of oracle-style validation within our ablation studies (Section 5.3.2 and Figure 10). To evaluate the fidelity of our SVD-based scoring proxy, we compared CADET against an oracle variant that performs an exhaustive enumeration of all possible rank assignments (up to a search limit of 20) within a restricted space.
>
> To provide a granular view of this alignment, we have added Table 3, which reports the detailed statistics comparing the proxy-estimated network sizes, the final decomposition sizes, and the absolute best sizes discovered via exhaustive rank enumeration. The data reveals that the constraint-solving estimated sizes are marginally conservative (slightly larger) but remain very close to the final decomposition sizes.
>
> Furthermore, when comparing end-to-end performance, the empirical results in Figure 10 demonstrate that CADET’s proxy-guided search closely tracks the oracle, yielding final compression ratios that are nearly identical to the exhaustive variant. Crucially, while the exhaustive enumeration requires more than two orders of magnitude more time to execute, CADET identifies these high-quality, near-optimal structures in approximately 10 seconds, validating both the efficiency and the accuracy of our proxy. We also evaluate where the structures selected by the exhaustive enumeration fall within CADET's cost-sorted candidate ranking, and the best tree structures from exhaustive enumeration consistently rank from 6 to 14 among more than 700 candidates.
>
> > Q3: Update the related work: The Tensor Network Structure Search discussion appears to stop at 2024. The authors should discuss relevant 2025-2026 work, or explain why it is not directly comparable.
>
> We thank the reviewer for helping us keep the paper up to date. We have expanded our related work discussion in Section 1.2 to include recent 2025-2026 advancements. Furthermore, we have integrated RGTN as an active baseline in our empirical evaluations (Figures 8 and 9). Our experiments indicate that while RGTN is thorough, it is prone to out-of-memory (OOM) errors when initialized with large tensor dimensions or applied to high-order tensors; when it does successfully run, it requires a high number of iterations to converge to the target reconstruction error if the error bound is tight.
>
> Another recent work, HISS, is not directly comparable as its algorithm is built atop the core contributions of this work.
>
> > Q4: Other minor request changes
>
> We have addressed and resolved all of the minor editorial and formatting requests in the revised manuscript.

---

### Review · Reviewer_iAYx · 2026-06-14

**Summary Of Contributions:**

This paper studies tensor network structure search for compressing multi-dimensional data tensors under a prescribed reconstruction error. The main idea is to decouple topology search from rank assignment. To do so, the authors introduce canonical dimension trees, which represent tree tensor network skeletons through nested index partitions while avoiding redundant and provably suboptimal structures. They then propose a fast scoring mechanism based on precomputing singular values for all feasible tensor matricizations. This turns rank assignment and candidate evaluation into a constraint-solving problem, avoiding full tensor decompositions for every candidate structure. The resulting system, Cadet, is evaluated on synthetic tensors and real datasets including light field data, BigEarthNet, and PDEBench.

The paper’s strengths are its clear algorithmic motivation, the clean separation between topology enumeration and rank optimization, the useful theoretical justification for the singular-value-based upper bound, and the empirical evidence showing favorable compression/time tradeoffs relative to several tensor network structure search baselines. The ablations also support the usefulness of both canonical enumeration and constraint-based rank search.

The main weaknesses are that the method is restricted to tree tensor networks, while some baselines can search cyclic structures; the scalability is still exponential in the tensor order due to the preprocessing over index bipartitions; and some claims about “optimal” or “near-optimal” structures should be more carefully scoped to the enumerated tree search space and the proposed conservative scoring proxy. The baseline comparison and reproducibility details would also benefit from more explicit hyperparameter, timeout, and implementation information.

**Audience:**

Yes

**Audience Explanation:**

This paper should be of interest to researchers working on tensor decompositions, tensor network compression, scientific machine learning, and efficient representation of high-dimensional data. The problem of choosing an appropriate tensor network structure is practically important, and the proposed decoupling of topology search and rank assignment is a useful algorithmic contribution. The paper is also relevant to readers interested in search-space canonicalization, efficient model selection under compression constraints, and scalable alternatives to repeated tensor decompositions during structure search. While the audience may be specialized, the work addresses a meaningful bottleneck in tensor network structure search and provides both theoretical and empirical support for a practical method.

**Broader Impact Concerns:**

I do not see major broader impact concerns that would require a dedicated expanded discussion.

**Claims And Evidence:**

Yes

**Claims Explanation:**

Overall, the main claims are supported, with some qualifications. The theoretical component justifies the use of singular values of the original tensor as a conservative proxy for singular values encountered during hierarchical decompositions, which supports the proposed constraint-based rank assignment procedure. The empirical evaluation covers synthetic data, several real datasets, comparisons against TT, HT, GreedyTN, TnALE, and SVDinsTN, and ablations isolating canonical enumeration and constraint-based rank search. These experiments provide convincing evidence that Cadet can find compressed tree tensor network structures efficiently and that its two main components contribute meaningfully to the result.

However, I think some claims should be stated more precisely. In particular, the method searches tree tensor networks only, so comparisons against methods that can discover cyclic structures should be framed with this scope in mind. Similarly, the terms “optimal” and “near-optimal” should be clarified: the scoring procedure provides a conservative upper-bound-based proxy, and the final result is best understood as the best structure found within the enumerated candidate class and search budget, rather than a global optimum over all tensor network structures. The scalability claims are also supported for the tested settings, but the paper should more clearly acknowledge that preprocessing all matricizations is exponential in tensor order and that the experiments mainly demonstrate practical feasibility up to moderate-order tensors.

**Requested Changes:**

1. As stated above, clarify the scope of the main claims, and qualify “optimal” and “near-optimal” terminology.
2. Add a clearer runtime and memory breakdown. Since the method relies on precomputing singular values for all feasible tensor matricizations, it would be helpful to separately report preprocessing time, enumeration/scoring time, final decomposition time, and memory usage for each dataset. This would make the practical scalability claims easier to assess.
3. Strengthen the discussion of scalability with tensor order. The complexity analysis already indicates exponential dependence on the number of tensor dimensions, but the main text should more explicitly discuss when the method is expected to become impractical and what settings are currently supported. The limitation should be connected to the empirical regime, where the method appears most practical for moderate-order tensors.
4. Clarify the theoretical guarantee and its assumptions. The singular-value upper-bound argument is important to the method. The paper should make the conditions under which the reconstruction error bound is guaranteed fully explicit, including the role of exact/truncated SVD, orthonormalization between decomposition steps, numerical precision, and the distinction between proxy cost and actual final compressed size.

---

> ### Author Response · Authors · 2026-07-04
> **Response to Reviewer iAYx**
>
> > Q1:  As stated above, clarify the scope of the main claims, and qualify “optimal” and “near-optimal” terminology.
>
> We have revised our claims regarding compression ratios in the manuscript including the Abstract and the Evaluation sections. Specifically, we have reframed the discussion to clarify that while methods exploring cyclic topologies invest significant computational resources to search a larger structural space, CADET serves as a highly efficient alternative. It achieves *comparable* compression ratios while reducing the search time by orders of magnitude.
>
> We have also qualified our terminology regarding “optimal” or “near-optimal” structures to prevent ambiguity. We now use the phrase "best/optimal tree structures within the search space" to explicitly distinguish our results from the optimum within the space that includes cyclic structures. Regarding the rank assignments, the constraint-solving results are empirically near-optimal given a specific dimension tree, as demonstrated in Section 5.3 (Table 3 and Figure 10); we have therefore kept this specific phrasing unchanged but ensured its context is precisely bounded.
>
> > Q2: Add a clearer runtime and memory breakdown. Since the method relies on precomputing singular values for all feasible tensor matricizations, it would be helpful to separately report preprocessing time, enumeration/scoring time, final decomposition time, and memory usage for each dataset. This would make the practical scalability claims easier to assess.
>
> We agree that a detailed computational breakdown enhances the transparency of our scalability claims. In the revised manuscript, we have included a comprehensive breakdown of preprocessing time, enumeration/scoring time, final decomposition time, and memory usage for each dataset in Table 1. As expected, the preprocessing phase dominates the total execution time and scales exponentially with the tensor order. We have added a detailed analysis of these performance trends in Section 5.1.2 to help readers better assess the practical scalability of CADET.
>
> > Q3: Strengthen the discussion of scalability with tensor order. The complexity analysis already indicates exponential dependence on the number of tensor dimensions, but the main text should more explicitly discuss when the method is expected to become impractical and what settings are currently supported. The limitation should be connected to the empirical regime.
>
> We explicitly address the exponential complexity relative to tensor order in two key sections of the revision:
> - Section 4: We have added an explicit discussion of the exponential growth term and contextualized why our empirical evaluation focuses on tensors up to order 6. This establishes a clear, practical regime where the algorithm remains highly effective.
> - Section 5.1.2 & Table 1: The newly added runtime breakdown in Table 1 empirically illustrates how preprocessing time scales with both tensor order and dimension size, providing a grounded view of the method's practical limits.
>
> > Q4: Clarify the theoretical guarantee and its assumptions. The singular-value upper-bound argument is important to the method. The paper should make the conditions under which the reconstruction error bound is guaranteed fully explicit, including the role of exact/truncated SVD, orthonormalization between decomposition steps, numerical precision, and the distinction between proxy cost and actual final compressed size.
>
> To clarify, the reconstruction error bound is strictly guaranteed without relying on restrictive external assumptions. Because the precomputed singular values serve as a rigorous upper bound for the true singular values, the tensor truncation performed over them is inherently conservative (the conservative scoring is close to the final decomposition result, and also approaches the best result from exhaustive search. These data are detailed in Table 3). Consequently, any rank assignment that satisfies the error bound for the upper-bound singular values is mathematically guaranteed to satisfy it for the true singular values.
>
> We have also clarified the specific roles of the underlying matrix operations and costs:
> - Truncated SVD & Orthonormalization: Truncated SVD is employed during the final decomposition to reduce bond dimensions, while sequential orthonormalization ensures that local truncation errors propagate correctly to the global error bound.
> - Constraint Solving Scores vs. Final Size: The objective function optimized by the constraint solver represents a conservative proxy cost. Because this proxy can overestimate the required resources, we employ a rounding step (detailed in Section 3.4) to reclaim additional compression opportunities in practice. The gap between the scoring results and the final sizes are compared in Table 3.
> - Numerical Precision: Standard standard-precision floating-point arithmetic is assumed for the SVD implementations, which we treat as a standard underlying system property.

---

### Review · Reviewer_wBLq · 2026-06-22

**Summary Of Contributions:**

This paper addresses the tensor network structure search (TN-SS) problem: given a data tensor and error tolerance ε, find the most compressed tree-structured tensor network. The core idea is to decouple topology enumeration from rank search, which prior work tackles jointly via expensive sampling. The contribution has two parts:
1. A representation of tensor network skeletons as rooted trees of index subsets, with pruning rules that eliminate duplicate and suboptimal topologies. The paper shows that canonicalization reduces the candidate pool from ~35,000 to 63 trees on a representative case.

2. Rather than running SVD-based decomposition on every candidate, the algorithm precomputes singular values for all tensor matricizations into a metadata map $\Omega$, then solves a small integer program per candidate that minimizes network size subject to cumulative truncation error ≤  $\epsilon$. Theorem 3.5 proves $\Omega$ singular values upper-bound those at any decomposition stage, so the ranking is conservative but sound.

**Audience:**

Yes

**Audience Explanation:**

Tensor network methods have a substantial and growing audience in ML, tensorized layers, model compression, probabilistic modeling, and scientific ML all depend on efficient low-rank tensor representations.

**Broader Impact Concerns:**

None.

**Claims And Evidence:**

Yes

**Claims Explanation:**

The empirical case is strong. Fig. 7 confirms the algorithm recovers or exceeds the ground truth in every case. Table 1shows reconstruction errors staying within ε across all datasets and tolerances, validating the soundness of the $\Omega$ upper bound. Fig. 8's comparison against five baselines across four datasets and two ε values is thorough, and the hyperparameter sweep in Fig. 9 addresses the concern that baseline tuning could explain the gap. The ablations cleanly separate the contributions of canonical enumeration and constraint-based scoring.

**Requested Changes:**

1. Decompose Table 1 times into $\Omega$ construction, tree enumeration, constraint solving, and final decomposition. If $\Omega$ construction dominates, the speedup claims need contextualizing.
2. Report $\Omega$ memory usage per dataset. Storing singular values for 2^{d-1} matricizations of a 6D tensor could be large.
3. Plot preprocessing time vs. d on fixed-mode-size synthetic tensors so readers know where the ceiling is. The complexity section gives O(n^{1.5d} 2^d) but a wall-clock number is more useful.
4. The equal-error baseline beating constraint solving on light field needs a sentence of explanation. If the solver is suboptimal in some regime, readers should know why and when.
5. Confirm all baselines ran on the same Apple M3 Max hardware. The TnALE OOM and SVDinsTN timeout claims are less meaningful if baselines ran on different machines.

---

> ### Author Response · Authors · 2026-07-04
> **Response to Reviewer wBLq**
>
> > Q1: Decompose Table 1 times into $\Omega$ construction, tree enumeration, constraint solving, and final decomposition. If  $\Omega$ construction dominates, the speedup claims need contextualizing.
>
> We counted the preprocessing time as part of the total search time when making claims about the speedup and plotting the figures. To make it clear, we have updated Table 1 to include a runtime breakdown across all four requested phases: $\Omega$ matrix construction (preprocessing), tree enumeration, constraint solving, and final decomposition. As the reviewer anticipated, the preprocessing phase ($\Omega$ construction) dominates the total execution time for higher-order tensors. We have added to Section 5.1.2 the analysis of the time domination at various data scales: CADET shifts the computational burden away from the structural search phase by performing a one-time, less expensive precomputation step. The overall speedup is achieved because we completely eliminate the need for repeated, expensive tensor decompositions during the topology and rank search, which heavily burdens iterative baseline methods.
>
> > Q2: Report memory usage per dataset. Storing singular values for 2^{d-1} matricizations of a 6D tensor could be large.
>
> We have added the empirical memory footprint for each dataset to Table 1. While storing the singular values for all $2^{d-1}$ matricizations scales with the tensor order $d$, our empirical results show that the memory usage for a 6D tensor remains well within the practical limits of standard modern workstations (e.g., peaking at 26GB for our largest datasets), demonstrating the feasibility of the approach in the target empirical regime.
>
> > Q3: Plot preprocessing time vs. d on fixed-mode-size synthetic tensors so readers know where the ceiling is. The complexity section gives O(n^{1.5d} 2^d) but a wall-clock number is more useful.
>
> We agree that wall-clock numbers provide highly practical context for understanding the scalability ceiling of the preprocessing phase. We have added a detailed analysis of these trends to the main text, primarily anchored by the empirical data in Table 1. The real-world benchmarks in Table 1 clearly demonstrate how preprocessing time scales exponentially with both tensor order and size: the smallest dataset (Light Field, 5th-order, ~600 KB) requires negligible preprocessing time; the medium dataset (BigEarthNet-Small, 4th-order, ~20 MB) requires several seconds; and the high-order datasets (BigEarthNet-Large and PDEBench, 6th-order, 1–2 GB) require thousands of seconds.
>
> To fully address the reviewer's request, we have also generated the suggested plot using synthetic data to isolate the relationship between tensor order ($d$), mode size ($n$), and preprocessing time. We have placed this plot in https://docs.google.com/document/d/e/2PACX-1vQi7By3L0DYM85w9Wu52wY7cV0PYydPeM_3a3gMu58lW5LxnJlJZYwodvtf8ze5vAqy9SvxMWTGJ_k8/pub. We also include this analysis in the appendix.
>
>
>
> > Q4: The equal-error baseline beating constraint solving on light field needs a sentence of explanation. If the solver is suboptimal in some regime, readers should know why and when.
>
> We have added an explicit explanation of this behavior to the result analysis in Section 5.3. The equal-error ablation baseline operates directly on the exact singular values of the tensor. Conversely, CADET utilizes precomputed, overapproximated upper-bound singular values to maintain its global error guarantee. Because these upper bounds are inherently conservative, CADET may occasionally select a slightly more conservative rank assignment or network structure in certain regimes (such as the Light Field dataset), leading to a marginally lower compression ratio than the equal-error baseline. We note that the performance gap remains minor and is a direct trade-off for CADET's strict theoretical guarantees.
>
> > Q5: Confirm all baselines ran on the same Apple M3 Max hardware. The TnALE OOM and SVDinsTN timeout claims are less meaningful if baselines ran on different machines.
>
> We confirm that all evaluations, including all baseline methods, were executed end-to-end on the exact same Apple M3 Max hardware configuration to ensure a perfectly controlled environment.
>
> We have corrected the typos in the text about the TnALE results: TnALE does not encounter an out-of-memory (OOM) error on the M3 Max hardware; rather, it fails to return any structural results within our strict timeout threshold when applied to large, high-order tensors. Other tools consistently hit the timeout threshold on this hardware. We have updated Section 5.2 to accurately describe these execution behaviors.